# Model updating of a wind turbine blade finite element Timoshenko beam model with invertible neural networks

**Pablo Noever-Castelos[1], David Melcher[2], and Claudio Balzani[1]**

[1]Leibniz University Hannover, Institute for Wind Energy Systems, Appelstr. 9A, Hanover, 30167, Germany
[2]Fraunhofer IWES, Fraunhofer Institute for Wind Energy Systems, Am Seedeich 45, 27572 Bremerhaven, Germany

**Correspondence:** Pablo Noever-Castelos (research@iwes.uni-hannover.de)

**Abstract.** Digitalization, especially in the form of a digital twin, is fast becoming a key instrument for the monitoring of a product's life cycle from manufacturing to operation and maintenance, and has recently been applied to wind turbine blades. Here, model updating plays an important role for digital twins, in the form of adjusting the model to best replicate the corresponding real-world counterpart. However, classical updating methods are generally limited to a reduced parameter space due to low computational efficiency. Moreover, these approaches most likely lack a probabilistic evaluation of the result.

The purpose of this paper is to extend a previous feasibility study to a finite element Timoshenko beam model of a full blade, for which the model updating process is conducted through the novel approach with invertible neural networks (INNs). This type of artificial neural network is trained to represent an inversion of the physical model, which in general is complex and non-linear. During the updating process, the inverse model is evaluated based on the target model's modal responses, which then returns the posterior prediction for the input parameters. In advance, a global sensitivity study will reduce the parameter space to a significant subset, on which the updating process will focus.

The finally trained INN excellently predicts the input parameters' posterior distributions of the proposed generic updating problem. Moreover, intrinsic model ambiguities, such as material densities of two closely located laminates, are correctly captured. A robustness analysis with noisy response reveals a few sensitive parameters, though most can still be recovered with equal accuracy. And, finally, after the resimulation analysis with the updated model, the modal response perfectly matches the target values. Thus, we successfully confirmed that INNs offer an extraordinary capability for structural model updating of even more complex and larger models of wind turbine blades.

## 1   Introduction

Wind turbine blades are enormous composite structures exposed to extreme and harsh environmental conditions. Due to these circumstances, structural health or condition monitoring plays a critical role in reliably ensuring the endurance of the rotor blade. However, this raises the need for an accurate model representation of the structure as built. In this context, the digital twin is emerging as a powerful instrument (Grieves, 2019) for these monitoring systems during operational time, though it can already be involved in early stages of manufacturing (Sayer et al., 2020). The concept of model updating is central to achieving a digital product twin, for

example, by updating the preliminary blade design based on sensor responses from blade characterization tests. This process of model updating ensures that the current stage of the digital twin represents the blade as built.

### 1.1   Model Updating of Wind Turbine Blades

Model updating has grown in importance in light of digitalization of the wind turbine blades, however, it is only marginally explored in literature. Similar to other structural dynamic model updating applications (Sehgal and Kumar, 2016), the publications on rotor blade model updating typically follow metaheuristic optimization techniques and de-

fine the objective function based on the modal assurance criterion (MAC), which represents a common metric for the quantitative comparison of modal shapes (Pastor et al., 2012). Other related modal metrics can be found in Allemang (2003). The most recent publications, such as Hofmeister et al. (2019) and Bruns et al. (2019), apply classical metaheuristic optimization algorithms to update the model parameters and localize damage in a generic problem with a finite element beam blade model. These publications evaluate a global pattern search and compare it to evolutionary, particle swarm, and genetic optimization algorithms. The objective function is based upon the natural frequencies and the MAC value. Furthermore, the MAC and the coordinate modal assurance criterion (COMAC) is applied in the model updating process of a finite element shell model of a rotor blade conducted by Knebusch et al. (2020). That study aims to update the blade model of a built blade along with high-fidelity modal measurements and a gradient-based optimization approach. Another approach presented by Schröder et al. (2018) uses a two-stage metaheuristic optimization to detect damages and ice accretion on a rotor blade. A global optimization with a simulated quenching algorithm is followed by a local method (sequential quadratic programming) to minimize the objective function, consisting of natural frequencies and mode shapes. Omenzetter and Turnbull (2018) implemented metaheuristic optimization methods (fireflies and virus optimization) to detect damages in a finite element beam model of a blade and compare the performance to a simplified beam experiment. Other publications cover simplified model updating procedures of low-level wind turbine blade models (Velazquez and Swartz, 2015; Liu et al., 2012; Lin et al., 2018). While most of the referred contributions focus on the field of damage detection, the model updating conducted by Luczak et al. (2014) highlights the impact of a flexible support structure of the test setup of modern blades, which was also revealed by Knebusch et al. (2020).

### 1.2    Drawbacks of Current Updating Approaches

Most of the these publications encounter three major problems:

1. Due to the aforementioned computational effort, the studies have been restricted to simple models
2. They typically lack an efficient probabilistic approach to evaluate the uncertainty of the results
3. All approaches only address one particular state of the blade at a defined condition and not a generalized inverse model

The aforementioned approaches can be classified as deterministic and thus lead to results which are not necessarily the global optima. Therefore, these methodologies may require the process to be run several times to ensure the result validity (Schröder et al., 2018; Omenzetter and Turnbull, 2018). This is especially problematic, since metaheuristic optimization algorithms are computationally expensive due to their iterative model evaluation (Chopard and Tomassini, 2018). As a reference, Bruns et al. (2019) performed 500 iterations for two updating parameters and 1,500 iterations for five updating parameters, while in Omenzetter and Turnbull (2018) the firefly optimization of two update parameters required 157 iterations until convergence and the virus optimization 5,000 iterations. Newer model updating techniques involve probabilistic approaches such as a sensitivity-based method (Augustyn et al., 2020) or Bayesian optimization (Marwala et al., 2016). The latter is based on sampling techniques such as Markov Chain Monte Carlo to cover the complete parameter space. However, these approaches typically require even more model evaluations as stated in Patelli et al. (2017). There, a relatively simple model of a 3 degree-of-freedom mass-spring system demanded 12,000 samples for the Bayesian solution, which was approximately 10 times higher than for the sensitivity-based method. Iterations are always model dependent, but to give a reference for the real time consumption, the model generator used in this publication (Noever-Castelos et al., 2021a) takes on average approx. 80s on a single-core device for one iteration, i.e., model creation. And finally, from the model updating we obtain one solution of input parameters for a particular set of model response parameters. However, if the model response changes the whole optimization process has to be repeated. While in most applications a solution for a particular model is sufficient, an inverted model, which maps model responses to input parameters, can be beneficial, e.g., in quality management during serial production. This reveals a niche for an efficient method to invert the physical model, enabling a fast evaluation of the model states at any time.

### 1.3    Model Updating via Invertible Neural Networks

This research framework is based on Noever-Castelos et al. (2021a), a feasibility study on a first structural level of wind turbine blades. The research performs a model updating with *conditional invertible neural networks* (cINN) (Ardizzone et al., 2019b) for four selected cross-sections of a wind turbine blade. Noever-Castelos et al. (2021a) considers a set of material and layup parameters as updateable inputs and takes cross-sectional structural beam properties, such as stiffness and mass matrix, as model outputs to define the objective values. A sensitivity analysis following a one-at-a-time approach identified a parameter subspace selection of 19 significant input parameters for the updating process. The specific objective of this current investigation in contrast to the aforementioned publication (Noever-Castelos et al., 2021a) is to:

1. Extend the feasibility study and methodology to a complete three-dimensional finite element Tymoshenko beam model of a wind turbine blade as applied in real

world problems, instead of analyzing isolated cross-sections

2. Introduce parameter splines for the input variation along the blade

3. Use modal blade shapes and frequencies as model response

4. Replace the sensitivity analysis for the parameter subspace selection by the global variance-based Sobol method (Sobol', 1993), which takes interactions of input parameters into account

5. Implement a pre-processing feed-forward neural network for the cINN conditions

6. Analyze the potential of replacing or neglecting the sensitivity analysis by training the cINN on the full parameter space

However, this investigation is still designed to reveal the feasibility with respect to a complex full three-dimensional Tymoshenko beam model, before applying the method to a high dimensional real-world and non-generic problem.

## 1.4   Outline

This study will follow the outline of Noever-Castelos et al. (2021a). The first section after the introduction presents the sensitivity analysis procedure and discusses the physical model built in MoCA (Model Creation and Analysis Tool for Wind Turbine Rotor Blades) (Noever-Castelos et al., 2021b) and BECAS (BEam Cross-section Analysis Software) (Blasques, 2012). The chosen architecture of the cINN is explained in Sect. 3. Sect. 4 covers the results discussion, with a general analysis of the updating results in Sect. 4.1. Sect. 4.2 reveals intrinsic model ambiguities, before the model robustness to noisy model responses is examined in Sect. 4.3. A resimulation analysis to ensure the high updating quality is performed in Sect. 4.4. Sect. 4.5 presents a method to replace the computational expensive sensitivity analysis. This is then all followed by the conclusion in Sect. 5.

## 2   Sensitivity Analysis of Modal Responses of a Rotor Blade Finite Element Beam Model

Typically a physical model consists of several input parameters defining the model behavior. The model is then evaluated or simulations are performed, which yield a model response. However, not all input parameters are equally contributing to the particular model response. A sensitivity analysis helps to identify the most significant input parameters. It is an underestimated powerful tool to reduce the model dimensions without loosing significant information. Especially for model updating purposes this can make a huge difference in performance. This section will discuss the applied sensitivity method as well as the applied model and parameter subspace selection.

## 2.1   Sobol' Sensitivity Method

Noever-Castelos et al. (2021a) performed a sensitivity analysis to reflect how input distributions influence the output distribution's variance and mean value in order to identify relevant input and output features for the model updating process with the invertible neural network. There, a one-at-a-time approach is used, where values vary individually and their impact on the output is analysed. In contrast to Noever-Castelos et al. (2021a), this contribution will make use of a variance-based approach, called Sobol method, or Sobol index (Sobol', 1993, 2001). This method is widely used in research and is used here, as it also applies globally to non-linear models and analyzes interaction of input parameters on the model response. For a multivariate function $y = f(x_1, \ldots, x_n)$, Sobol derived the 1st order Sobol index $S_i$ for the variable $x_i$ as follows:

$$S_i = \frac{V\left[\mathbb{E}\left(y|x_i\right)\right]}{V\left(y\right)} \tag{1}$$

This is a measure to what extent the impact of varying $x_i$ will result on the output $y$. On the basis of a random sampling of the parameters $x$, $\mathbb{E}(y|x_i)$ represents the expectation $E$ of all $y$ for a constant value of $x_i$. It can be understood as an average of $y$ corresponding to a slice of the $x_i$ domain in the parameter space. $V\left[\mathbb{E}\left(y|x_i\right)\right]$ is then the variance of all expectations over the range of values of $x_i$, i.e., slices of the $x_i$ domain (Saltelli et al., 2008). This variance is finally related to the overall variance of $y$. The 1st order Sobol index ranges in $0 \leq S_i \leq 1$. Higher-order Sobol indices can also be extracted, see Saltelli et al. (2008), which measure the sensitivity of parameter interactions. For instance the 2nd order Sobol index shows the joint effect of two parameters on the output, whereas 3rd indices express the joint effect of three parameter interactions, and so on. Although these indices can give a significant inside into the model, such as existing collinearities, the number of indices grow geometrical with the number of parameters, which quickly makes the computation intractable. However, the total Sobol index $S_{Ti}$ gathers the total sensitivity for a parameter including the first order and all higher order interactions. According to Saltelli et al. (2008) the total index $S_{Ti}$ is calculated as follows:

$$S_{Ti} = 1 - \frac{V\left[\mathbb{E}\left(y|x_{\sim i}\right)\right]}{V\left(y\right)} \tag{2}$$

Where $V\left[\mathbb{E}\left(y|x_{\sim i}\right)\right]$ describes the variance of all expectations over the range where $x_i$ is not included. If the model is purely additive for a particular parameter, the corresponding total Sobol index should be equal to the 1st order index. While the total index does not provide the information of which interaction is significant, it does identify if any interaction exist, with the benefit, that it is computed alongside the 1st order Sobol index without any significant additional computational effort.

For a multivariate function with multiple outputs $(y_1, \ldots, y_m) = f(x_1, \ldots, x_n)$ Eq. (1) and Eq. (2) can be expressed, respectively, as:

$$S_{ij} = \frac{V\left[\mathbb{E}\left(y_j | x_i\right)\right]}{V\left(y_j\right)} \tag{3}$$

$$S_{Tij} = 1 - \frac{V\left[\mathbb{E}\left(y_j | x_{\sim i}\right)\right]}{V\left(y_j\right)} \tag{4}$$

## 2.2  Rotor Blade Finite Element Beam Model

The necessary model generation and variation is performed with the model creator MoCA (Noever-Castelos et al.,
2021b) and its interface to BECAS (Blasques and Stolpe, 2012) to create cross-sectional beam properties, which are assembled to a finite element beam (FE beam) and evaluated in ANSYS Mechanical (ANSYS Inc., 2021b). We will be performing the analysis on the DemoBlade of the Smart-
Blades2 project (SmartBlades2, 2016-2020). Figure 1 depicts a coarse version of the FE beam used in this study. In contrast to this simplified visualisation in Fig 1, the applied FE beam model is built of 50 3D linear beam elements (BEAM188) (ANSYS Inc., 2021a) with higher mesh density
to the root and tip section of the blade, where greater geometrical and material changes are expected. Thus, the finite element model consists of 51 nodes ($N_{\mathrm{FE}}$). The input parameter selection of Noever-Castelos et al. (2021a) was slightly expanded to cover more material properties, which will be
discussed in detail later. The input parameter selection spans a space with a maximum dimension of $\overline{\mathbb{D}}_{CS} = 33$ for each cross-section, though varying these for each of the 50 cross-sections would result in $\mathbb{D}_{tot} = 1,650$ parameters. Assuming a smooth variation of each parameter over the radius, Akima
splines (Akima, 1970) were introduced to represent the parameter variation along the blade. An exemplary spline is depicted in Fig. 2. The spline is built based upon five equidistant nodes, that may vary in $y$-direction within the given variation range of the parameter. The number of spline nodes
can be chosen arbitrary; however, a high number increases the computational costs (more updating parameters) and can lead to collinear behavior if the nodes are to near, whereas a low number reduces the flexibility to adapt to short distance changes. For this study the number where chosen based on
experience as a trade-off between computational costs and a sufficient approximation of a global parameter variation.

Table 1 summarizes all the investigated input parameters $x_i$ and corresponding properties. Moreover, Table 1 lists the number of spline nodes with their respective normalized
radial range and variance limits for each property. In this feasibility study, we consider the most significant independent elastic properties for each material: the density $\rho$, the Young's modulus $E_{11}$, the shear modulus $G_{12}$, and the Poisson's ratio $\nu_{12}$, which may be varied over all five nodes in

a range of $\pm 10\%$. Here, we have neglected all thickness-
50 related elastic constants, i.e., parameters including the index/direction 3 and $E_{22}$, as these parameters offer no significant contribution to the stiffness terms of the beam cross-sectional properties according to Hodges (2006) and Noever-Castelos et al. (2021a). Since foam is modeled as an isotropic
material, only two independent elastic properties $E$, $G$ and the density $\rho$ are considered. In addition to the material properties, the division points are also varied. These subdivide the shell in cross-sectional direction into different sections with a constant material layup or define sub-component po-
sitions such as the web or adhesive (Noever-Castelos et al., 2021b). The division point parameters $P$ are allowed to vary on the three mid nodes by the given absolute range. The root and tip node cannot be varied due to model generation issues within MoCA, thus the variance for node $N_0$ and $N_4$
will be kept at zero, similar to Fig. 2. All applied variations are approximately twice the permitted manufacturing tolerances (Noever-Castelos et al., 2021a), in order to assure some flexibility of the inverse model. Summing up all parameters and nodes, the problem spans a parameter space
of $\dim(x) = 153$. The sensitivity study is conducted based on the Python package $SALib$ (Herman and Usher, 2017) and a random sampling dimension of $n = 2^9 = 512$ samples. $SALib$ uses the quasi-random sampling with low-discrepancy sequences technique from Saltelli et al. (2008) for the sensi-
tivity analysis. To compute the Sobol index, the algorithms require a variation of each input feature individually for each of the $n$ samples, which results in a total sample size of $n_{\mathrm{total}} \cdot (\dim(x) + 2) = 79,360$ to compute the 1st and total order Sobol indices. The sensitivity study as well as the up-
dating process is based on the modal beam response $y$, as described in Gundlach and Govers (2019), in a free-free and a clamped scenario, which is comparable to an elastic suspension of the blade and a fixation of its root to a test rig, respectively. In each case, the first 10 eigenmodes are ex-
tracted, excluding the rigid body motion modes in the free-free scenario. For all 10 mode shapes of each configuration (free-free and clamped), the natural frequency and the three deflections and three rotations of each finite element beam node $N_{\mathrm{FE}}$ are saved. These are collected in a response matrix
with $\dim(y) = (10+10) \cdot (1+6) = 140$ columns. Throughout this paper, input parameters and model responses will also be referred to as input and output features or conditions, respectively.

## 2.3  Feature Subspace Selection with Sobol Indicies

After computing the 1st order and total Sobol index $S_{ij}$ and $S_{Tij}$ respectively for each input feature $x_i$ and output feature $y_i$ at every $N_{\mathrm{FE}}$ position, we obtain a matrix of size 140 x 51 x 153, i.e., $\dim(y)$ x $\dim(N_{\mathrm{FE}})$ x $\dim(x)$. For the subspace selection we follow two selection method:

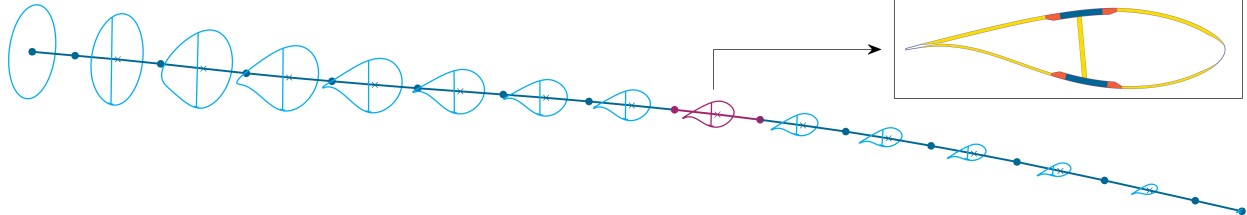

**Figure 1.** Exemplary finite element beam with reduced number of elements and exemplary cross-sectional illustration. The detail shows a cross-sectional BECAS output (Blasques and Stolpe, 2012) as used in the feasibility study (Noever-Castelos et al., 2021a).

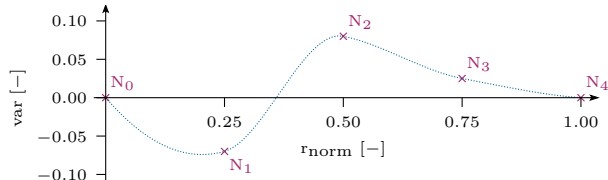

**Figure 2.** Exemplary variation spline with five nodes.

**Table 1.** Input feature list analyzed in this study. Each feature and property builds a distribution spline based on the given number of equidistant nodes within the given normalized radial range of the blade. Each node value may then vary in the listed variance range.

| Parameter | Property | Nodes | Norm. range | Variance |
|---|---|---|---|---|
| UD | $\rho, E_{11}, G_{12}, \nu_{12}$ | 5 | [0, 1] | $\pm 10\%$ |
| Biax45° | $\rho, E_{11}, G_{12}, \nu_{12}$ | 5 | [0, 1] | $\pm 10\%$ |
| Biax90° | $\rho, E_{11}, G_{12}, \nu_{12}$ | 5 | [0, 1] | $\pm 10\%$ |
| Triax | $\rho, E_{11}, G_{12}, \nu_{12}$ | 5 | [0, 1] | $\pm 10\%$ |
| Flange | $\rho, E_{11}, G_{12}, \nu_{12}$ | 5 | [0, 0.1] | $\pm 10\%$ |
| Balsa | $\rho, E_{11}, G_{12}, \nu_{12}$ | 5 | [0, 1] | $\pm 10\%$ |
| Foam | $\rho, E, G$ | 5 | [0, 1] | $\pm 10\%$ |
| $P_{SS,TE,offset}$ | Location | 3 | [0.25, 0.75] | $\pm 10$ mm |
| $P_{SS,Mid,spar\,cap}$ | Location | 3 | [0.25, 0.75] | $\pm 15$ mm |
| $P_{SS,LE,offset}$ | Location | 3 | [0.25, 0.75] | $\pm 10$ mm |
| $P_{PS,TE,offset}$ | Location | 3 | [0.25, 0.75] | $\pm 10$ mm |
| $P_{PS,Mid,spar\,cap}$ | Location | 3 | [0.25, 0.75] | $\pm 15$ mm |
| $P_{PS,LE,offset}$ | Location | 3 | [0.25, 0.75] | $\pm 10$ mm |

1. By computing the maximum appearing $1^{st}$ order Sobol index of each input feature and comparing it to a threshold
2. By performing a singular value decomposition (SVD) on the total Sobol sensitivit matrix to idetifiy the most relevant contributions and mapping these back onto the input feature with a QR factorization with column pivoting (Chakroborty and Saha, 2010; Olufsen and Ottesen, 2013)

The selected subspaces are merged to a final subspace, which is applied for the model updating process.

For the first selection method the sensitivity matrix containing the $1^{st}$ order Sobol index is condensed to a single maximum value $S_{\max\,i}$ for each input feature $x_i$. Therefore, it is reduced to identify relevant input features $y$ by computing the maximum value along the other non-corresponding dimensions, i.e., dimension 2 and 3. Subsequently, an arbitrary threshold $S_{thld}$ is defined to reject all features with a lower maximum index $S_{\max\,i}$. By this, we aim to consider only features which have a significant impact during at least one event at one location, thus containing enough information for the updating process. Based on experience, we have chosen $S_{thld} = 0.1$.

The second method follows a combination of SVD and QR factorization on the sensitivity matrix of the total Sobol index according to Chakroborty and Saha (2010) for a given set of $n$ input parameters $x$. Here each mode shape is analyzed individually. Therefore, the sensitivity matrix is divided and reshaped; the $1^{st}$ dimension, i.e., the 6 DOFs plus frequency and $2^{nd}$ dimension, i.e., the node positions $N_{FE}$, are flattened, while the $3^{rd}$ dimension, i.e., input features, is kept yielding a $(m \times n)$ matrix. Given this individual total Sobol sensitivity matrix $\mathbf{S_T}$ for each mode shape the singular value decomposition according to Golub and van Loan (2013) is:

$$\mathbf{S_T} = \mathbf{U\,\Sigma\,V^T} \tag{5}$$

$\mathbf{U}$ and $\mathbf{V}$ denote the left and right singular vector matrices, each column corresponding to the singular values in $\mathbf{\Sigma} = \mathrm{diag}\{s_1, s_2, \cdots, s_p\}$ with $p = \min(m, n)$, respectively. According to Chakroborty and Saha (2010), the criterion *percentage of energy explained by the singular values* is used to identify the $g$ most relevant features. The percentage of energy $P_{ex}$ is calculated as normalized cumulative sum of the singular values:

$$P_{ex} = \frac{\sum_{i=1}^{g} s_i^2}{\sum_{i=1}^{p} s_i^2} \cdot 100\% \tag{6}$$

The number of relevant singular values $g$ is equal to the highest number $g$ complying $P_{ex} \leq 99\%$. The rest $p - g$ singular

values only contribute to 1% of the total energy and are therefore insignificant for the result.

A subsequent QR-factorization with column pivoting (Golub and van Loan, 2013) is used according to Olufsen and Ottesen (2013); Chakroborty and Saha (2010) to extract the order of the original input vector $x$, by sorting the columns of the left singular vector matrix $\mathbf{V}$ of size $n \times n$ in order of maximum Euclidean norm in successive orthogonal directions:

$$\mathbf{V^T}\,\mathbf{P} = \mathbf{Q}\,\mathbf{R} \tag{7}$$

$\mathbf{Q}$ is a matrix with orthonormal columns, R is an upper triangular matrix and P is the permutation matrix. In this particular case of a square matrix $\mathbf{V}$, all matrices are of the same dimension as $\mathbf{V}$. The permutation matrix $\mathbf{P}$ is finally applied to the input parameter vector $x$ to resort the vector according to sensitivity significance:

$$x_{\mathrm{s}} = x^T \cdot \mathbf{P} \tag{8}$$

The sorted input vector $x_s$ is than reduced to the first $g$ entries, representing the most significant parameters for the analyzed mode shape following the criterion explained above. After computing all $x_s$ for each mode shape these are all merged to a final set of input parameters determined to be relevant during at least one mode shape. With this SVD-QR method applied to the total Sobol indices matrix, the authors tried to identify parameters that are significant either on their on or in interaction with others. However, the significance is not measured as maximum value in one occasion, such as in the first method, but rather contribute substantially on average over a complete mode shape.

Both methods lead to the 49 selected features depicted in Table 2 with their respective $S_{\mathrm{max}}$ and a checkmark showing the selection by the SVD-QR method.

When analyzing the rejections, it has to be noted that all structural properties are condensed to cross-sectional beam properties. That means, for example, Biax 45° as a face layer of the shear web is typically located near the elastic and gravitational center of the cross-sections and thus does not contribute in excess to the mass inertia according to the Steiner theorem, nor to the overall bending stiffness (Gross et al., 2012). Consequently, a variation of $\rho_{Biax45}$ and $E_{11,\mathrm{Biax45}}$ will not significantly impact the modal response of the beam model. However, its shear modulus $G_{12,\mathrm{Biax45}}$ does have an impact when dealing with the shear forces from flap-wise loading. Regarding foam and balsa as sandwich core materials, the stiffness contribution to the sandwich panels is approximately 1% compared to the GFRP (glass fiber-reinforced plastic) face sheets and this makes their variations neglectable, while the mass contributions depending on the layup can reach up to $66 - 100\%$, which is why a few of the density spline nodes are kept. Summarizing the sensitivity analysis reduced the input feature space

to $\dim(x_{\mathrm{sel}}) = 45$, approximately 30% of $\dim(x)$. The output features were all kept according to the feature selection approach. However, a reduced set of radial positions can be applicable as the intrinsic information might be repeated in neighboring $N_{\mathrm{FE}}$. This repeated information does not necessarily improve the updating results, but reduce the computational performance. Therefore, the output of each third node is selected, ending up with a radial output dimension of $\dim(N_{\mathrm{FE,sel}}) = 17$. Thus, the final dimension for the model updating process of the input feature space is $\dim(input) = \dim(x_{\mathrm{sel}}) = 45$ and of the output feature space is $\dim(output) = \dim(N_{\mathrm{FE,sel}})$ x $\dim(y) = 17$ x $140$.

**Table 2.** Selected feature list from sensitivity study with their respective maximum $1^{st}$ order Sobol indicies $S_{\max i}$ (values shown in bold meet the given threshold $S_{thld} = 0.1$) and the selection mark for the SVD-QR method.

| Feature | $S_{\max i}$ | SVD | Feature | $S_{\max i}$ | SVD | Feature | $S_{\max i}$ | SVD | Feature | $S_{\max i}$ | SVD | Feature | $S_{\max i}$ | SVD |
|---|---|---|---|---|---|---|---|---|---|---|---|---|---|---|
| $\rho_{UD,N1}$ | **0.248** | ✓ | $G_{12,Biax45,N3}$ | **0.149** | ✓ | $\rho_{Triax,N3}$ | **0.462** | ✓ | $G_{12,Triax,N3}$ | **0.593** | ✓ | $G_{12,Flange,N3}$ | 0.015 | ✓ |
| $\rho_{UD,N2}$ | **0.381** | ✓ | $\rho_{Biax90,N3}$ | **0.240** | ✓ | $\rho_{Triax,N4}$ | **0.804** | ✓ | $G_{12,Triax,N4}$ | **0.343** | ✓ | $\rho_{Balsa,N1}$ | **0.207** | ✓ |
| $\rho_{UD,N3}$ | **0.278** | ✓ | $\rho_{Biax90,N4}$ | **0.109** |  | $E_{11,Triax,N0}$ | **0.312** | ✓ | $\rho_{Flange,N0}$ | **0.214** |  | $\rho_{Foam,N2}$ | **0.162** | ✓ |
| $E_{11,UD,N0}$ | **0.109** | ✓ | $E_{11,Biax90,N0}$ | **0.116** | ✓ | $E_{11,Triax,N1}$ | **0.375** | ✓ | $\rho_{Flange,N1}$ | **0.620** | ✓ | $P_{SS,Mid,spar\,cap,N0}$ | **0.669** | ✓ |
| $E_{11,UD,N1}$ | **0.434** | ✓ | $E_{11,Biax90,N1}$ | **0.142** |  | $E_{11,Triax,N2}$ | **0.485** | ✓ | $E_{11,Flange,N0}$ | 0.087 | ✓ | $P_{SS,Mid,spar\,cap,N1}$ | **0.433** | ✓ |
| $E_{11,UD,N2}$ | **0.432** | ✓ | $E_{11,Biax90,N2}$ | **0.132** | ✓ | $E_{11,Triax,N3}$ | **0.351** | ✓ | $E_{11,Flange,N1}$ | **0.413** | ✓ | $P_{SS,Mid,spar\,cap,N2}$ | **0.458** | ✓ |
| $E_{11,UD,N3}$ | **0.371** | ✓ | $E_{11,Biax90,N3}$ | **0.218** | ✓ | $E_{11,Triax,N4}$ | **0.527** | ✓ | $E_{11,Flange,N2}$ | **0.485** | ✓ | $P_{PS,Mid,spar\,cap,N0}$ | **0.491** | ✓ |
| $G_{12,Biax45,N0}$ | 0.099 | ✓ | $G_{12,Biax90,N3}$ | **0.112** | ✓ | $G_{12,Triax,N0}$ | **0.371** | ✓ | $E_{11,Flange,N3}$ | 0.044 | ✓ | $P_{PS,Mid,spar\,cap,N1}$ | **0.549** | ✓ |
| $G_{12,Biax45,N1}$ | **0.224** | ✓ | $\rho_{Triax,N1}$ | **0.291** | ✓ | $G_{12,Triax,N1}$ | **0.607** | ✓ | $G_{12,Flange,N1}$ | **0.331** | ✓ | $P_{PS,Mid,spar\,cap,N2}$ | **0.433** | ✓ |
| $G_{12,Biax45,N2}$ | **0.262** | ✓ | $\rho_{Triax,N2}$ | **0.211** | ✓ | $G_{12,Triax,N2}$ | **0.521** | ✓ | $G_{12,Flange,N2}$ | **0.275** | ✓ |  |  |  |

## 3 Invertible Neural Network Architecture

Before proceeding to the model updating process, it is necessary to define the *invertible neural network architecture*. Similar to Noever-Castelos et al. (2021a), this work will built on *conditional invertible neural networks* (cINN) (Ardizzone et al., 2019b) implemented in FrEIA – Framework for Easily Invertible Architectures (Visual Learning Lab Heidelberg, 2021). A basic cINN consists of a sequence of *conditional coupling blocks* (CC), as shown in Fig. 3. Each of these represents affine transformations that can easily be inverted. The embedded *sub-networks* $s_1$, $t_1$, $s_2$, $t_2$ embody the trainable functions of this type of artificial neural network.

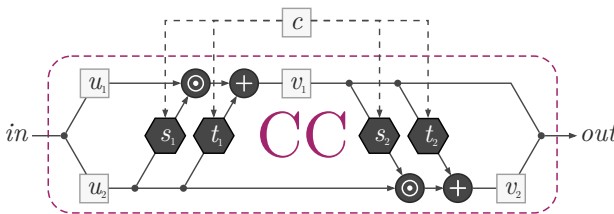

**Figure 3.** The *conditional coupling blocks* CC with its embedded *sub-network* $s_1, t_1, s_2, t_2$. This CC architecture can easily be inverted. Ardizzone et al. (2019b)

These *sub-networks* stack the conditions $c$ and the input slice $u_2$ or $v_1$ and transform them for further processing. The stacking necessarily requires similar spacial dimensions of $c$ and $u_2$ or $v_1$, respectively. For a further brief introduction to cINN with topic-related application, please refer to Noever-Castelos et al. (2021a). A more in-depth explanation can also be found in Ardizzone et al. (2019b) and Ardizzone et al. (2018).

After an extensive hyperparameter study, the presented investigation applies the network depicted in Fig. 4. Hyperparameters describe the network or architecture parameters of artificial neural networks, like number of layers or perceptrons. It consists of a cINN (blue) with a sequence of 15 CCs, grouped into clusters of three. This cINN transforms between the beam input $x$ and the latent space $z$. However, unlike the underlying feasibility study Noever-Castelos et al. (2021a), an additional feedforward network is implemented, referred to as a *conditional network* (orange). The idea is to preprocess the raw conditions $c$, i.e., beam responses, before passing them to the *sub-networks* in the CCs. It is trained in conjunction with the cINN, to extract relevant feature information optimally for each stage. The *conditional network* architecture is inspired by Ardizzone et al. (2019b) and should extract higher-level features of $c$ to feed into the sequential CCs, which, according to Ardizzone et al. (2019b), should relieve the sub-networks from having to relearn these higher-level features each time again. With a conditional beam response $c$ of shape $\dim(c) = \dim(N_{FE,sel})$ x $\dim(y)$, the *conditional network* applies 1D-convolutions (conv 1D) to process the data, which gradually increase in size to progressively extract higher-level features, which are fed into the different clusters of the cINN.

In general, the beam input would also be available in a 2D shape (property $\times$ spline nodes), though the feature selection of the sensitivity analysis reduced the splines irregularly. Thus, a 2D shape cannot be maintained anymore, as not all splines have the same number of nodes. Therefore, the selected beam input $x$ for the updating process going into the cINN, is flattened to a vector and is not present in a 2D shape, as for example the beam response $c$. A consequence is, that the sub-networks cannot make use of convolutional layers, but have to include feed-forward layers. However, this will not have any significant impact on the result. As mentioned before, the conditions and input features are stacked in the *sub-networks*, which thus need a similar spacial shape. Consequently, the *conditional network* has to flatten the shape to a vector for each output, in order to agree with the input shape in the sub-networks. Before flattening the output, the *conditional network* activates the convolutional layer output with a *parametric rectified linear unit* (PReLU) (He et al.,

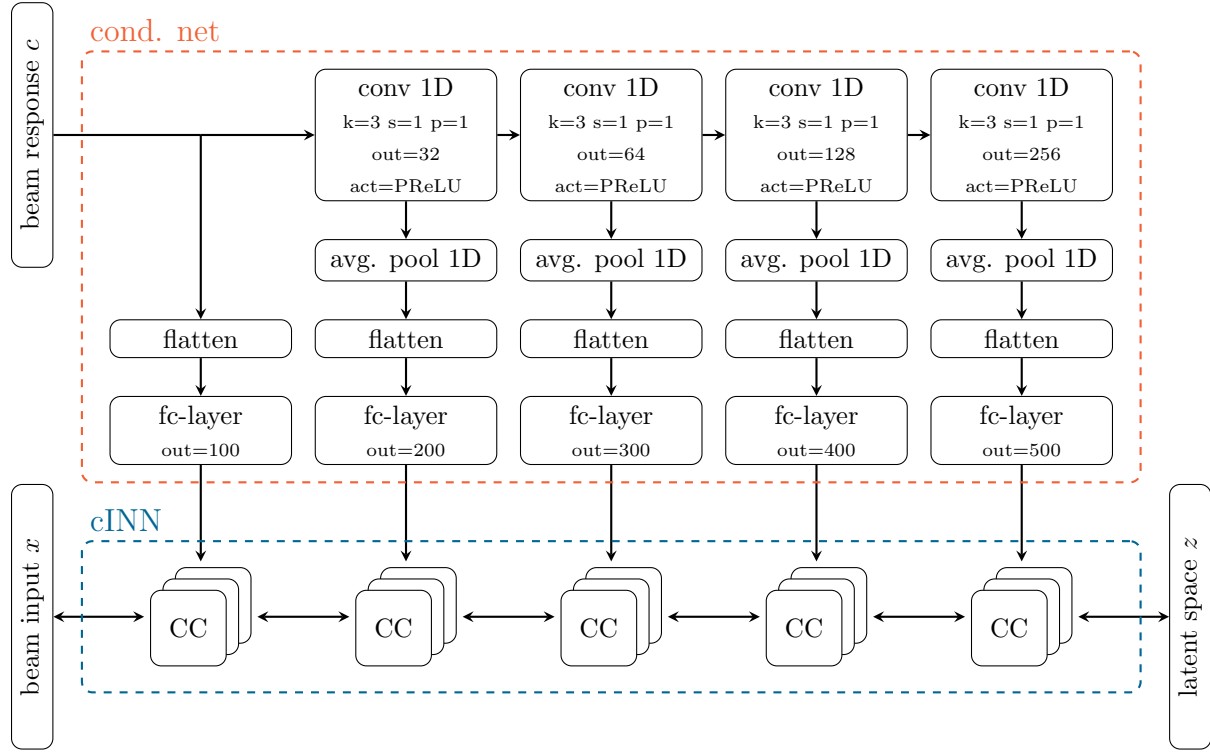

**Figure 4.** *Conditional invertible neural networks* (cINN, in blue frame) with sequentially connected *conditional coupling blocks* CC. The conditional feed-forward network (cond. net, in orange frame) preprocesses the condition $y$ with 1D convolutional layers and PReLU (parametric rectified linear unit) activations. Average 1D pooling is performed on the output, before it is flattened and reduced in dimension with a fully connected layer (fc-layer), to be then fed into the *sub-networks* of the CCs. The convolutions gradually increases in size in order to progressively extract higher-level features from the condition $c$.

2015) and halves the dimension with an average 1D pooling layer (Chollet, 2018) (avg. pool 1D). After flattening, the dimension is additionally reduced with a fully connected layer (fc-layer) to subsequently relieve the sub-network's computation.

Within the cINN, the CCs are clustered into groups, which are then each fed by the progressively processed conditions $c$. All *sub-networks* have one hidden fc-layer, followed by a batch normalization to improve generalization and a PReLU (Chollet, 2018) activation layer, as depicted in Fig. 5. As previously explained the *conditional network* processes the conditions $c$ and has 5 outputs at different stages of the processing. Each of this outputs is fed into a cluster of 3 CCs. the configuration for each cluster and the corresponding hyperparameters for the *conditional network*, cINN and *sub-networks* is summarized in Table 3.

The training is performed with an AdaGrad optimizer (Duchi et al., 2011) and an initial learning rate of 0.3, which is gradually decreased throughout the training process. The optimization minimizes the negative logarithmic likelihood (NLL) given in Eq. (9) in order to match the model's posterior prediction of $p_x(x|y)$ with the true posterior of the inverse problem (Noever-Castelos et al., 2021a).

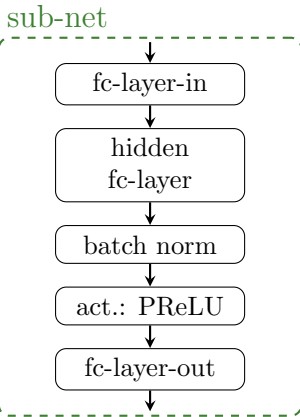

**Figure 5.** *Sub-network* with one hidden fully connected layer (fc-layer), batch normalization, and a PReLU activation layer. Each *conditional coupling blocks* CC has such a *sub-network* embedded.

$$\mathcal{L}_{\mathrm{NLL}} = \mathbb{E}\Big[-\log\big(p(x_i \mid y_i)\big)\Big]$$
$$= \mathbb{E}\left[\frac{\|f(x_i; y_i)\|^2}{2} - \log|\det(J_i)|\right] + const. \qquad (9)$$

**Table 3.** Hyperparameter set of the complete network, including *conditional network*, *conditional invertible neural networks* (cINN), and *sub-network*. The cINN is divided into 5 clusters, for which the hyperparameters are listed separately. In Cluster 1, the conditions are directly fed into the *conditional coupling blocks* CC, without a prior convolutional layer (cf. Fig. 4).

|  |  |  | Cluster 1 | Cluster 2 | Cluster 3 | Cluster 4 | Cluster 5 |
|---|---|---|---|---|---|---|---|
| Conditional network | Conv 1D | kernel $k$ |  | 3 | 3 | 3 | 3 |
|  |  | stride $s$ |  | 1 | 1 | 1 | 1 |
|  |  | padding $p$ |  | 1 | 1 | 1 | 1 |
|  |  | out chan. $out$ |  | 32 | 64 | 128 | 256 |
|  | Activation |  |  | PReLU | PReLU | PReLU | PReLU |
|  | Avgerage 1D pooling | kernel $k$ |  | 2 | 2 | 2 | 2 |
|  |  | stride $s$ |  | 2 | 2 | 2 | 2 |
|  |  | padding $p$ |  | 0 | 0 | 0 | 0 |
|  | Flatten |  | ✓ | ✓ | ✓ | ✓ | ✓ |
|  | Fully connected | nodes | 100 | 200 | 300 | 400 | 500 |
| cINN | Conditional coupling block (CC) |  | 3 | 3 | 3 | 3 | 3 |
| Sub-network | Fully connected | nodes | 400 | 500 | 600 | 700 | 800 |
|  | Batch normalization |  | ✓ | ✓ | ✓ | ✓ | ✓ |
|  | Activation |  | PReLU | PReLU | PReLU | PReLU | PReLU |

## 4   Model Updating of a Rotor Blade Beam Model

Having selected the significant features with the sensitivity analysis and defined the cINN architecture, we will now move on to the model updating process and its evaluation.
Therefore, the workflow of the cINN if briefly explained along with the schematic view of the transformations performed by the cINN in Fig. 6. The presented cINN in Sect. 3 is trained and tested with sample sets of input features $x$ and their corresponding conditions $c$ in the form of the modal
beam responses as described in Sect. 2. The concept and training of the cINN is based on the Bayes' theorem to infer a posterior distribution $p_x(x|c)$ from a set of conditions $c$. Therefore, the cINN learns the conditioned transformation from the posterior distribution $p_x(x|c)$ onto the latent dis-
tribution $p_z(z)$, as depicted in Fig. 6. This mapping can be achieved through maximum likelihood training. The training is performed over 150 epochs, i.e., training iterations, with a samples size of 30,000 training samples, in order to minimize the negative log-likelihood $\mathcal{L}_{\mathrm{NLL}}$ (given in Eq. (9)). For
a more detailed description of the inherent method of cINNs please refer to Noever-Castelos et al. (2021b) or Ardizzone et al. (2019a). Additionally a sample set of 5,000 test samples, which have not been seen by the cINN during its training, are used for validating and testing the cINN after the
training. All input features are always sampled randomly and independently, but at the same time, in order to span the complete parameter space. However, only features selected by the sensitivity study (cf. Table 2) are passed on to the cINN, as the other parameters are identified to be less relevant. As the
cINN is trained to map the input features $x$ into a normally distributed latent space $p_z(z)$, during the inverse evaluation the process is reversed: the latent space is sampled from a

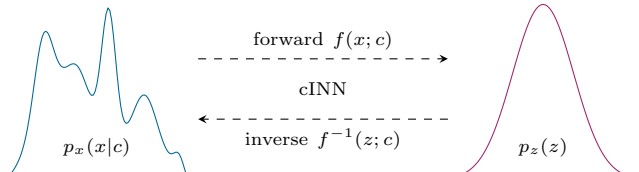

**Figure 6.** Schematic view of the transformation between the input features $x$ and the latent space $z$ for a given condition $c$ performed by the conditional invertible neural network. (Noever-Castelos et al., 2021a)

Gaussian normal distribution (e.g., 50-100 samples), which the cINN then transforms along with the beam response as condition $c$ to the posterior prediction of the input features. 35 This prediction results in a distribution for each input feature $p_x(x|y)$ as depicted in Fig. 6. In order to generalize the data for the training process and make it more comparable for the evaluation, all input features and conditions are standardized to zero mean and a standard deviation of 1 over the 40 complete training set. The necessary scaling factors are additionally saved in the cINN to transform back and forth any input features or conditions used in the cINN besides the training process. Consequently, all features and conditions during the evaluation of the cINN are related to the complete train- 45 ing set's mean and standard deviation. Generally, the posterior predictions are also depicted with respect to their ground truth, i.e., target value of the sample, to align multiple samples for enhanced comparison.

This section first analyses the overall updating results of 50 the model. The identified inference ambiguities are then highlighted and discussed, before the model is checked

against its robustness to noisy conditions $c_{\text{noisy}}$. Based on the predicted posterior distribution of the input features $p(x|y)$, a resimulation analysis is performed, where the updated parameters are used to feed the phyiscal model and calculate
the beam response, in order to check the quality of the updating and sensitivity analysis results. Finally, a method for avoiding the computational intensive sensitivity analysis is presented.

## 4.1 General Analysis of the Updating Results

In the first instance the posterior distributions have to be examined. Figure 7 shows as an example the predicted posterior distribution of four different input features as a histogram and fitted Gaussian distribution. The ground truth on the x-axis represents the real value used to generate the sample,
while the distribution is obtained from the cINN. For the further analysis, the type of distribution must be known in advance for it to be possible to apply the correct metric, e.g., mean or median. In this case of a Gaussian distribution, the mean is the most significant value and will thus be applied
in this study to reduce the posterior prediction distribution to a single value accompanied by the standard deviation as a measure of uncertainty.

By shifting the former x-axis from Fig. 7 onto the y-axis and reducing the distributions to their mean and standard de-
25 viation, as stated before, we obtain the graphs depicted in Fig. 8 for the same exemplary sample, but with all updated parameters. Most values range close to their ground truth value and with a narrow distribution, which is desired. For some input features, e.g., $\rho_{\text{Biax90,N4}}$, the prediction is less ac-
30 curate. However, the overall posterior prediction in this example is very good, as approx. 70% of the predictions are within a range of $\pm 0.05$ (standardized scale) of the ground truth.

After having checked the results in detail for one exem-
35 plary sample, Fig. 9 shows the prediction result of all selected input features for the 5,000 test samples. The graphs scatter the standardized mean posterior prediction $\bar{p}(x|y)$ against their corresponding target value from the sample set. Thus, the ideal case would correlate to an exact line with a slope
$m = 1$ and an interception $b = 0$. Each graph is equipped with the coefficient of determination $R^2$ and shows a corresponding regression line with slope $m$. Approximately 70% of the selected features reach a very satisfying linear correlation with $R^2 > 0.9$, while showing a slope $m$ of approx. 0.9
or higher. For the rest of the discussion the we will be sticking with the $R^2$-value for the accuracy, as the slope accuracy correlates with the $R^2$-value.

In the following we will create the link between the sensitivity results to enhance the comprehension and explain
possible discrepancies. In general a high sensitivity $S_{\text{max},i} > 0.35$ leads to a high prediction accuracy ($R^2 > 0.9$). A second major metric to fully understand the prediction accuracy is the cross-correlation of the input features, which re-

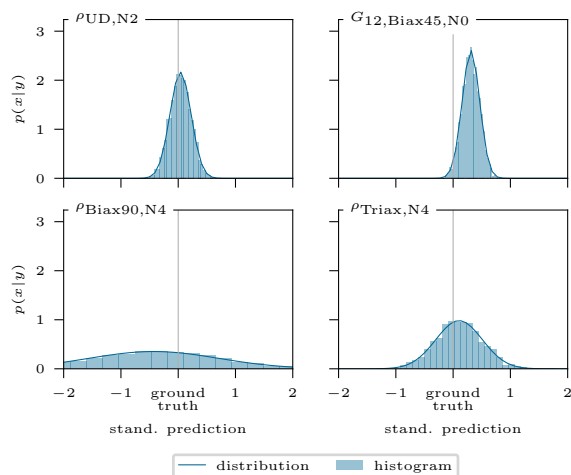

**Figure 7.** *Conditional invertible neural network's* standardized posterior prediction distributions $p(x|y)$ for four input features of one example. Plotted as a histogram and fitted Gaussian distribution.

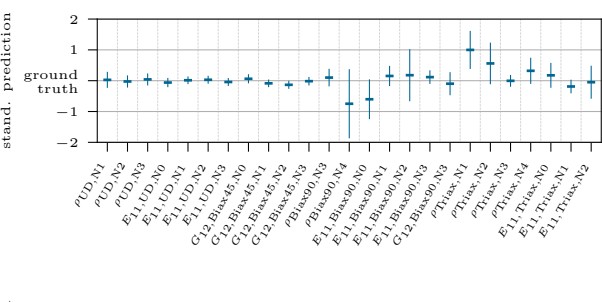

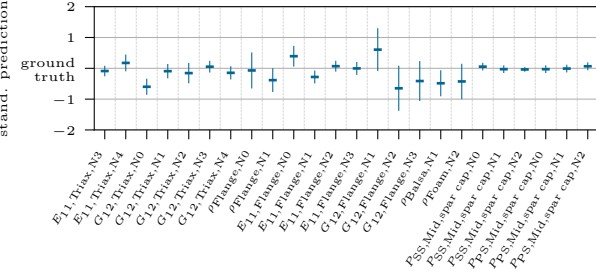

**Figure 8.** The first two graphs show the standardized posterior prediction for all updated input features related to the target value with 1-$\sigma$ standard deviation as error, thus the mean value marks the distance to the target value, i.e., ground truth.

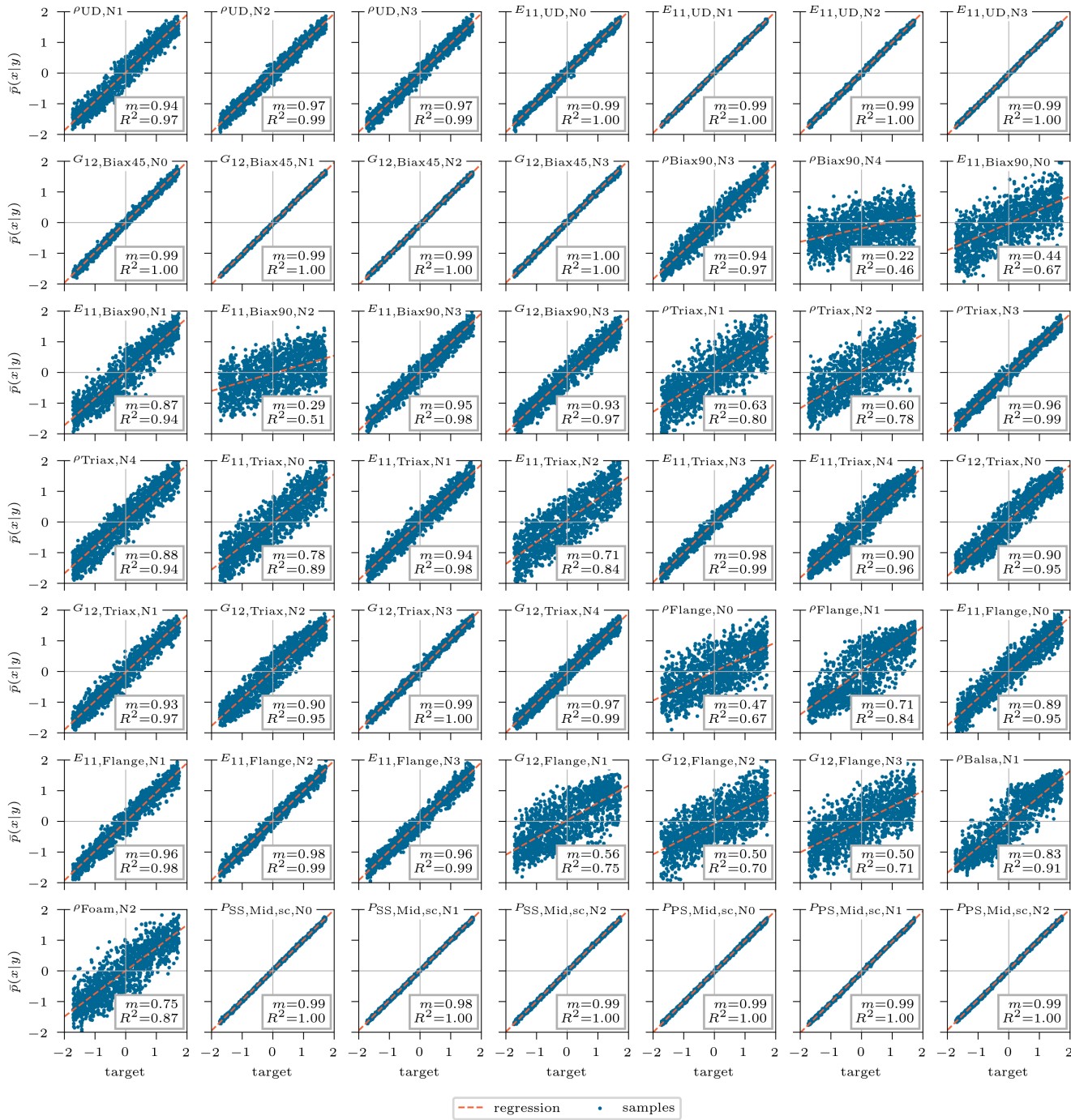

**Figure 9.** Standardized mean of posterior prediction $\bar{x}$ of the updated inputs over the corresponding target standardized value for the 5,000 test samples. The coefficient of determination and a corresponding linear regression line are shown. The corresponding parameter description to the features can be found in Table 2.

veals collinearities within the physical model. These present a problem for the inversion of the model, as the output response of the physical model is ambiguous and can be traced back to different combinations of input features. However, this will be addressed in Sect. 4.2. Input features that do not

have any substantial cross-correlation, but high $S_{\mathrm{max},i}$, reach prediction accuracies of $R^2 \approx 1.0$, e.g., all spar cap position points $P_{\mathrm{Mid,sc}}$ or the Young's modulus of the UD material $E_{11,\mathrm{UD}}$. For instance, $\rho_{\mathrm{Flange,N1}}$ has one of the highest sensitivity index $S_{\mathrm{max},i} = 0.62$, but a comparable poor prediction

accuracy of $R^2 = 0.82$. This fact is due to a strong collinearity to the input features $\rho_{\text{Flange,N0}}$, $\rho_{\text{Flange,N2}}$. In contrast, the feature $G_{12,\text{Biax45,N0}}$ has a low sensitivity $S_{\text{max},i} = 0.1$ but an excellent prediction accuracy of $R^2 = 1.0$. The reason is that this feature does not show any collinearity to other features. Although the $S_{\text{max},i}$ is low, according to the $1^{\text{st}}$ order Sobol index matrix it has at three nodes $N_{\text{FE}}$ the second and third highest contribution of all input features for a particular mode shape and DOF, reaching a magnitude of 50-75% of the maximum value for that DOF. That shows the powerful capability of the cINN to learn the mapping of an input feature to only a very few output features out of the complete response data. Table 4 completes this list of examples with the most striking discrepancies of sensitivity index to prediction accuracy of the input features. Hence, the sensitivity analysis is a good indication to detect a significant parameter subspace for the model updating, though high sensitivities do not directly promise highly accurate inverse prediction.

**Table 4.** Most striking discrepancies of sensitivity and prediction accuracy of input features.

| Feature | $S_{\max i}$ | $R^2$ | $\text{XCorr}_{min}$ | Explanation |
|---|---|---|---|---|
| $E_{11,\text{UD},\text{N0}}$ | 0.110 | 1.000 | -0.663 | Low $S_{\max i}$, however, for two sensors it has the third highest contribution in a DOF during one mode shape. The values reach a magnitude of 66% and 50% of the maximum value in their corresponding DOF. |
| $G_{12,\text{Biax45},\text{N0}}$ | 0.100 | 1.000 | -0.179 | Low $S_{\max i}$, however, for three sensors it has the second and third highest contribution in a DOF during one mode shape. The values reach a magnitude of 75%, 55% and 53% of the maximum value in their corresponding DOF. |
| $G_{12,\text{Biax45},\text{N3}}$ | 0.149 | 1.000 | -0.383 | Low $S_{\max i}$, however, for one sensor it has the third highest contribution in a DOF during one mode shape. The value reaches a magnitude of 83% of the maximum value in its corresponding DOF. |
| $\rho_{\text{Triax},\text{N1}}$ | 0.292 | 0.790 | -0.537 | Mid $S_{\max i}$; Mixed collinearity with $\rho_{\text{Biax90},\text{N1}}$ and $\rho_{\text{Balsa},\text{N1}}$ |
| $\rho_{\text{Triax},\text{N2}}$ | 0.211 | 0.770 | -0.678 | Mid $S_{\max i}$; Mixed collinearity with $\rho_{\text{Biax90},\text{N2}}$ and $\rho_{\text{Balsa},\text{N2}}$ |
| $\rho_{\text{Flange},\text{N0}}$ | 0.214 | 0.660 | -0.952 | Mid $S_{\max i}$; Strong colinearity with $\rho_{\text{Flange},\text{N1}}$ |
| $\rho_{\text{Flange},\text{N1}}$ | 0.620 | 0.820 | -0.952 | High $S_{\max i}$; Strong colinearity with $\rho_{\text{Flange},\text{N0}}$ and $\rho_{\text{Flange},\text{N2}}$ |
| $G_{12,\text{Flange},\text{N1}}$ | 0.332 | 0.720 | -0.857 | Mid $S_{\max i}$; Strong colinearity with $G_{12,\text{Flange},\text{N1}}$ |
| $G_{12,\text{Flange},\text{N2}}$ | 0.276 | 0.690 | -0.876 | Mid $S_{\max i}$; Strong colinearity with $G_{12,\text{Flange},\text{N0}}$ and $G_{12,\text{Flange},\text{N2}}$ |

## 4.2 Intrinsic Model Ambiguities

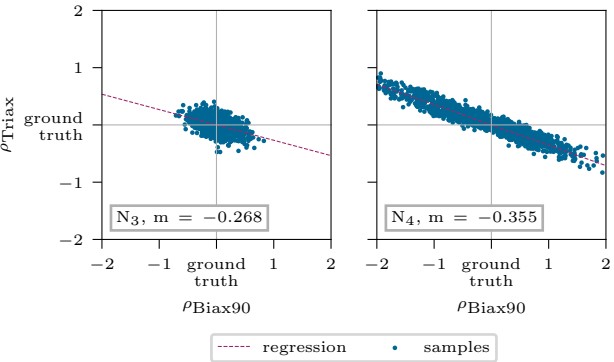

**Figure 10.** Interaction of density $\rho_{\text{Biax90}}$ and $\rho_{\text{Triax}}$ describing the intrinsic model ambiguities. The depicted values correspond to the standardized mean posterior prediction for the 5,000 test samples.

Ambiguities can originate from different sources, such as little significant responses or modeling issues (Ardizzone et al., 2019a). Noever-Castelos et al. (2021a) revealed some intrinsic model ambiguities of counteracting density values of the Biax90° and Triax layer in the blade cross-section. This was also handled by the cINN in this study, although it was only checked for the two spline nodes $N_3$ and $N_4$, as these coincide in the feature selection. The results are depicted in Fig. 10, showing the standardized mean posterior prediction for the 5,000 test samples related to their ground truth and the linear regression as well as the corresponding slope $m$ in the label. While the mean posterior predictions at node $N_3$ were detected rather accurately (cf. Fig. 8), i.e., represent a circular area in Fig. 10, the values of node $N_4$ spread more and correlate to the plotted regression line.

In addition to the density, another ambiguity was detected in the Young's modulus $E_{11}$ of both these materials, shown in Fig. 11 for the nodes $N_{0-3}$. Here, the correlation of the mean posterior predictions is reasonably well described by the calculated regression lines. Finally, the last correlation was found for the shear modulus $G_{12,\text{N3}}$ between the same materials (Fig. 12).

All ambiguities rely on the same fact that the Biax90° and Triax layers appear subsequently in the stacking of the sandwich panels of the blade shell. The stacking is schematically illustrated in Fig. 13 with a detailed view of the shell, showing the stacking in exploded view. Together, these layers build the symmetric inner and outer face sheets of the shell, with a layer thickness of $t_{\text{Biax90}} = 0.651\,mm$ and $t_{\text{Triax}} = 0.922 mm$, same density $\rho_{\text{Biax90}} = \rho_{\text{Triax}} = 1,875\,\frac{kg}{m^3}$, Young's modulus $E_{11,\text{Biax90}} = 26,430\,\frac{N}{mm^2}$ and $E_{11,\text{Triax}} = 29,873\,\frac{N}{mm^2}$, and shear modulus $G_{12,\text{Biax90}} = 3,464\,\frac{N}{mm^2}$ and $G_{12,\text{Triax}} = 6,918\,\frac{N}{mm^2}$.

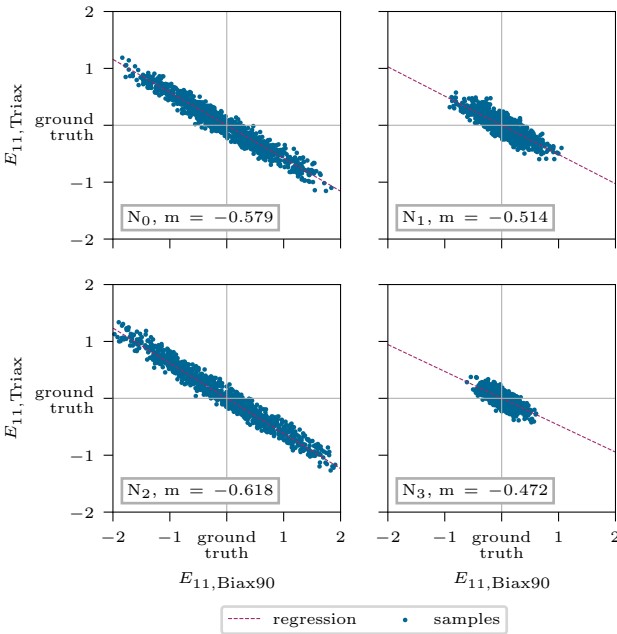

**Figure 11.** Interaction of stiffness $E_{11,\text{Biax90}}$ and $E_{11,\text{Triax}}$ describing the intrinsic model ambiguities. The depicted values correspond to the standardized mean posterior prediction for the 5,000 test samples.

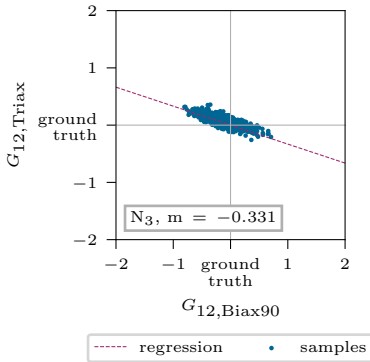

**Figure 12.** Interaction of shear stiffness $G_{12,\text{Biax90}}$ and $G_{12,\text{Triax}}$ describing the intrinsic model ambiguities. The depicted values correspond to the standardized mean posterior prediction for the 5,000 test samples.

The contributions of the properties to the model behavior must be analyzed for it to be possible to understand these ambiguities further. As described in Sect. 2, a finite element beam model is composed of beam elements containing cross-5 sectional properties (Blasques, 2012). These basically consist of mass and stiffness terms, which can be directly linked to $\rho$ and $E_{11}$ or $G_{12}$, respectively (Hodges, 2006). The upcoming deductions follow classical mechanics theories found for example in Gross et al. (2012). First, considering the mass 10 contribution, we stick with the simplified example of the cen-

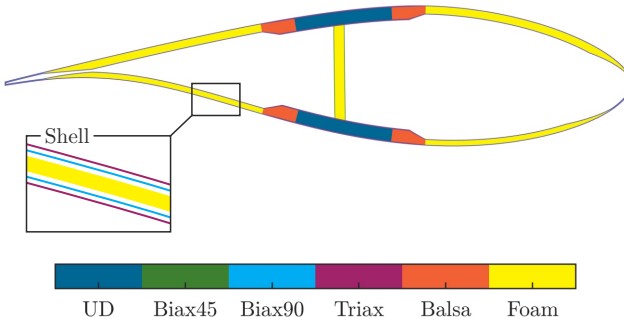

**Figure 13.** Schematic blade cross-sectional view at a radial position of $r = 12$ m with a detailed explosion drawing of the shell.

ter of gravity:

$$x_s = \frac{1}{m_{tot}} \int x^2 dm = \frac{1}{m_{tot}} \sum x_j^2 m_j \qquad (10)$$

where $x_j$ represents the center of gravity of each component and $m_i$ the corresponding mass. Due to the very thin thickness of both layers and the overall cross-sectional dimension 15 being about $10^3$ greater for both materials, it can be assumed that $x_j = x_s$. And by expecting that the cINN correctly predicts the total mass $m_{tot}$, Eq. (10) yields:

$$x_s = \frac{1}{m_{tot}} \cdot x_s \sum m_j \qquad (11)$$

$$m_{tot} = \sum m_j \qquad (12)$$

$$= k_{\text{Biax90}} \cdot t_{\text{Biax90}} \cdot \rho_{\text{Biax90}} + k_{\text{Triax}} \cdot t_{\text{Triax}} \cdot \rho_{\text{Triax}} \qquad (13)$$

And this obviously leads to the summation of all individual masses to the total mass, where $k$ represents the number of layers. This of course holds for higher order moments of mass, due to the given proximity of both layers. Thus, a ratio 25 between both materials can be expressed:

$$k_{\text{Biax90}} \cdot t_{\text{Biax90}} \cdot \rho_{\text{Biax90}} \; : \; k_{\text{Triax}} \cdot t_{\text{Triax}} \cdot \rho_{\text{Triax}} \qquad (14)$$

A similar behavior is also found for the stiffness. This is explained in a simplified example for the flexural rigidity of a beam in Eq. (15), which extends with the Steiner theorem to 30 Eq. (16).

$$EI_{\bar{x}} = \sum E_j I_{\bar{x},j} \qquad (15)$$

$$= \sum E_j (I_{x,j} + x_s^2 \cdot A_j) \qquad (16)$$

Assuming the layers have a rectangular shape, the area moment of inertia is $I_{x,j} = \frac{w \cdot t^3}{12}$, though the width $w$ of the layer 35 is large, the thickness $t$ is $10^{-3}$ smaller and thus this term vanishes. With that, Eq. (16) reduces to Eq. (17). As stated before, $x_s$ can be assumed to be constant and the same holds for the width $w_i$, as in the cross-sectional direction both material layers cover the complete circumference of the blade. 40

This results in the proportionality in Eq. (18)

$$EI_{\bar{x}} = \sum E_j(x_s^2 \cdot A_j) = \sum E_j(x_s^2 \cdot k_j \cdot t_j \cdot w_j) \qquad (17)$$

$$EI_{\bar{x}} \propto \sum E_j \cdot k_j \cdot t_j \qquad (18)$$

Similarly, to the total mass $m_{tot}$, we expect the cINN to predict the global $EI_{\bar{x}}$ accurately and, consequently we can establish the following ratio for the stiffness:

$$k_{\text{Biax90}} \cdot t_{\text{Biax90}} \cdot E_{\text{Biax90}} \; : \; k_{\text{Triax}} \cdot t_{\text{Triax}} \cdot E_{\text{Triax}} \qquad (19)$$

Analog derivations can be made for the shear modulus, which ends up in the ratio:

$$k_{\text{Biax90}} \cdot t_{\text{Biax90}} \cdot G_{\text{Biax90}} \; : \; k_{\text{Triax}} \cdot t_{\text{Triax}} \cdot G_{\text{Triax}} \qquad (20)$$

Figure 14 shows the number of each layer for the respective material along the blade, which corresponds to both the inner and outer face sheet of the shell. The corresponding spline nodes positions are also depicted. Table 5 shows the ratios according to Eq. (14), Eq. (19), and Eq. (20) of the different possible stacking options in Fig. 14. Looking back to

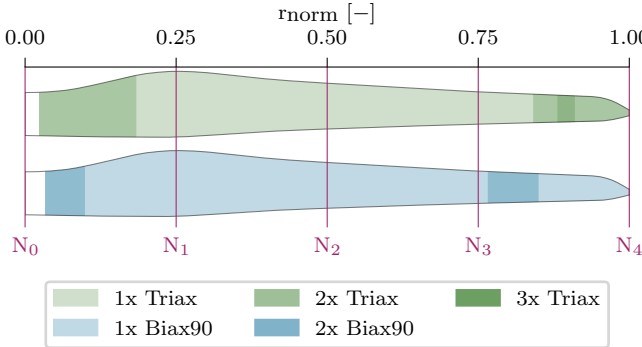

**Figure 14.** Layup of the sandwich laminate face sheets of the blade shell, consisting of Triax and Biax90°. The inner and outer face sheets are symmetric.

**Table 5.** Ratio between Biax90° and Triax layers for density and stiffness contribution, considering different layer constellation.

| $k_{\text{Biax90}}$ | 1 | 1 | 1 | 2 | 2 | 2 |
|---|---|---|---|---|---|---|
| $k_{\text{Triax}}$ | 1 | 2 | 3 | 1 | 2 | 3 |
| $\frac{k_{\text{Biax90}} \cdot \rho_{\text{Biax90}} \cdot t_{\text{Biax90}}}{k_{\text{Triax}} \cdot \rho_{\text{Triax}} \cdot t_{\text{Triax}}}$ | $\frac{0.706}{1}$ | $\frac{0.353}{1}$ | $\frac{0.235}{1}$ | $\frac{1.412}{1}$ | $\frac{0.706}{1}$ | $\frac{0.471}{1}$ |
| $\frac{k_{\text{Biax90}} \cdot E_{\text{Biax90}} \cdot t_{\text{Biax90}}}{k_{\text{Triax}} \cdot E_{\text{Triax}} \cdot t_{\text{Triax}}}$ | $\frac{0.625}{1}$ | $\frac{0.312}{1}$ | $\frac{0.208}{1}$ | $\frac{1.249}{1}$ | $\frac{0.625}{1}$ | $\frac{0.416}{1}$ |
| $\frac{k_{\text{Biax90}} \cdot G_{\text{Biax90}} \cdot t_{\text{Biax90}}}{k_{\text{Triax}} \cdot G_{\text{Triax}} \cdot t_{\text{Triax}}}$ | $\frac{0.354}{1}$ | $\frac{0.177}{1}$ | $\frac{0.118}{1}$ | $\frac{0.707}{1}$ | $\frac{0.354}{1}$ | $\frac{0.236}{1}$ |

the identified ambiguities in Fig. 10 of the density at node $N_4$, the linear regression shows a slope of $m = -0.355$. Assuming each spline node contributes to the variance of half of the space to the left and right of it, the given slope agrees

extremely well with the ratio of $k_{\text{Biax90}} = 1$ and $k_{\text{Triax}} = 2$. This corresponds to the stacking shown near the node $N_4$ in Fig. 14. Due to the poor linear regression of node $N_3$ in Fig. 10, the slope is not reliable, thus no conclusion can be drawn.

However, the counteracting Young's moduli in Fig. 11 can be very accurately captured by the ratios for most spline nodes. Starting with Node $N_2$ (figure 11 bottom-left), which is clearly affected by only one layer to the left and right of it (cf. Fig. 14), the line slope $m = -0.618$ matches the value in Table 5 ($k_{\text{Biax90}} = 1$, $k_{\text{Triax}} = 1$) of 0.625. Node $N_0$ has a slope of $m = -0.579$, which agrees well with the value corresponding to $k_{\text{Biax90}} = 2$ and $k_{\text{Triax}} = 2$, but tending towards $k_{\text{Biax90}} = 1$ and $k_{\text{Triax}} = 2$, which is also in the scope of this node according to the layup in Fig. 14. Similar behavior is found for node $N_1$. Node $N_3$ does not fully agree with this argumentation, though the point scatters less and the regression line might not be accurate enough. The same holds for the shear modulus in Fig. 12.

As assumed in the derivation of the ratios, we can state that the cINN should correctly predict the total mass and the stiffness contributions in a global manner, but suffers from an intrinsic model ambiguity affected by the counteracting densities $\rho$, Young's moduli $E_{11}$, and shear moduli $G_{12}$ of the neighbouring materials Biax90° and Triax. However, it offers posterior predictions for these features, but with a wide distribution expressing the uncertainty of the cINN based on the given ambiguity. Merging both materials to a face sheet material following laminate theory, would avoid these ambiguities and improve the prediction qualities for the overall laminate. It is assumed that, based on the relatively low layer thickness, the infusion and therefore the fiber volume fraction of both layers is very similar, so that this approach should be valid.

### 4.3 Model Robustness

So far the analysis of this feasibility study was conducted on the exact test sample data, i.e., for a given input sample the corresponding exact output sample is generated with the tool chain MoCA + BECAS + ANSYS. In future studies, this presented method should be applied to real measured data of a blade and this normally suffers from measurement uncertainties. It is thus important to analyze the model robustness with respect to a measurement error of the output features. Therefore, an error of $5\%$ as normally distributed random noise is applied to the clean output response of each sample, which is then used as a condition to infer the posterior prediction of the input features. The results are shown in Fig. A1 in the appendix, comparing the noisy (orange) and the clean (blue) mean posterior predictions $\bar{p}(x|y)$ against their corresponding targets for all 5,000 test samples. The graphs show some features that are sensitive for noise, such as $E_{11,\text{Flange,N0-3}}$, $G_{12,\text{Flange,N3}}$. As visually confirmed in Fig. A1, the other features do not show a wider spread (orange) than the original

values (blue) and therefore do not suffer from any accuracy loss. Additionally, tests were performed resuming the training of the cINN with noisy conditions in order to improve the prediction quality, though no benefit was identified.

## 4.4 Resimulation Analysis

A resimulation analysis aims to utilize the posterior predictions of the cINN based on the original response to resimulate/recalculate the response with the physical model in order to compare it to the original response used to perform the prediction. For all samples, the correct input features and their corresponding response features are known, which we will be referring to as targets. The target response is used as a condition for the cINN to infer the posterior prediction of the selected input features. From these inferred input features we can create new input splines for each input, as depicted exemplarily in Fig. 15. However, the prediction is not a discrete value but a Gaussian distribution as we have seen before in Sect. 4.1. Additionally, there are nodes that the sensitivity analysis excluded from the updating process; these may take every value within their variation range, as they were sampled uniformly. Hence, for each input feature we obtain a range of possible splines as Fig. 15 illustrates. Here, the orange spline represents the target variance of the input parameter and the dark blue area represents the expected value, i.e., the mean prediction from the updated nodes. In the case of the first spline for $\rho_{UD}$, nodes $N_0$ and $N_4$ were excluded from the updating process and can thus take any value in the range of $\pm 10\%$, as we do not have any prediction for them. As such the blue area covers all possible splines a user would take as the result from the model updating process. However, the purpose of this first evaluation of the resimulation analysis is to examine, if sampling splines from the given distributions will all lead to appropriate results. Therefore, the $1-\sigma-$uncertainty displayed in light blue shows the standard deviation of the predicted nodes. In this first analysis, we sample uniformly from the not updated nodes (dark blue range) and normally distributed from the updated nodes (light blue) to create a spline. This will be done 1,000 times for the same given target response of the selected single test sample. Subsequently, these 1,000 sets of input splines are then used to create the model and calculate its modal response. For the sake of completeness, Table A2 gathers the identified mode shapes of both configurations. The resultant mode shapes of the free-free and the clamped configurations are then compared to the target response with the help of the modal assurance criterion (MAC) (Allemang, 2003).

$$\text{MAC}_{ij} = \frac{|\Phi_i \cdot \Phi_j|^2}{|\Phi_i \cdot \Phi_i| \cdot |\Phi_j \cdot \Phi_j|} \tag{21}$$

The MAC is the scalar product of two normalized vectors, each representing all the model's degrees of freedoms of a particular mode shape. It is basically an orthogonality check: equal mode shapes reach a value of MAC $= 1$, while a value

of MAC $> 0.8$ is already assumed to show good coherence (Pastor et al., 2012). For a multiple number of modes, a MAC matrix summarizes all MAC values of all mode shapes compared against each other.

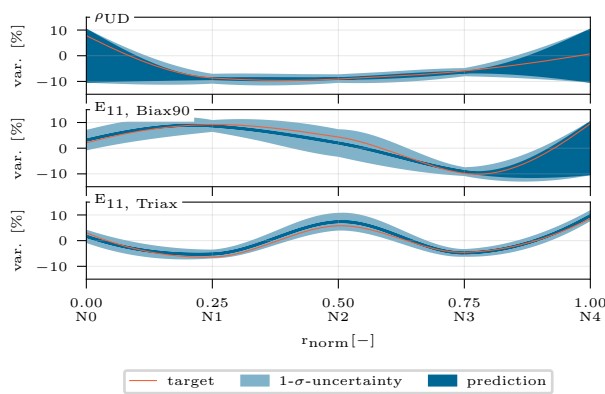

**Figure 15.** Exemplary inferred spline prediction range for $\rho_{UD}$, $E_{11,\text{Biax90}}$ and $E_{11,\text{Triax}}$. The graphs depict the target spline in orange, the mean prediction in dark blue, and the $1-\sigma$-uncertainty in light blue, for the updated spline nodes.

In our use case, the MAC matrix is computed individually for all responses of the previously generated 1,000 samples against the target response. For the free-free configuration, Fig. 16 illustrates the mean value of the MAC matrix over all samples in the top graph. The corresponding standard deviation is depicted below. The main diagonal ideally takes values of $\text{MAC}_{ij} = 1$, as the same mode shape of the sample and the target is compared. Additionally, the matrix should be symmetric, as the comparison of $\text{MAC}_{ij} = \text{MAC}_{ji}$ represents the same two mode shapes. Figure 16 confirms this ideal symmetric matrix structure for the re-simulated samples, with mean values $\overline{\text{MAC}} > 0.9975$ in the diagonal and extremely low standard deviations of $\sigma_{\text{MAC}} < 0.003$. For the clamped configuration, the values on the diagonal are also strikingly close to one ($\overline{\text{MAC}} > 0.9960$, $\sigma_{\text{MAC}} < 0.005$) and the overall matrix appears symmetric. In this way, sampling from the distribution predicted by the cINN for each selected input feature and arbitrarily choosing a value for the not updated values yields an exact coherence of target and computed mode shapes.

After having analysed a single target sample, the resimulation is expanded to more samples to show the cINN's general performance. Therefore, posterior predictions for the 5,000 test samples of the test set are inferred with the cINN. Contrary to the resimulation case before, only one input is generated for each of the samples by choosing the mean value of the prediction and, in the case of excluded variables, a node value of zero (i.e., no variation). That represents a typical choice a user would make, based on predicted posterior distributions. Figure 17 depicts the mean (horizontal marker),

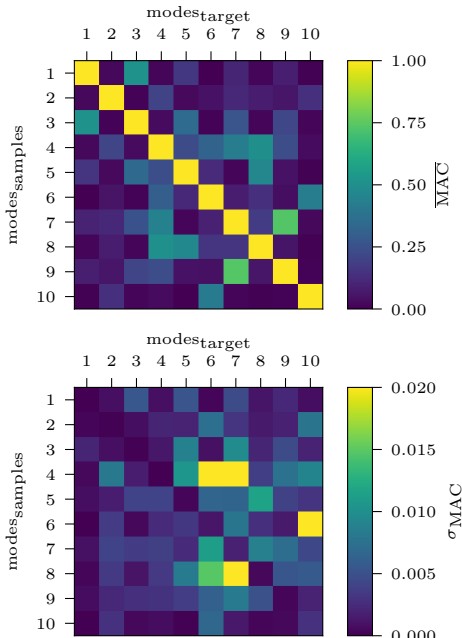

**Figure 16.** Mean values (top) and standard deviations (bottom) of the MAC matrix for the free-free modal configuration based on 1,000 spline samples inferred for one target response.

max and min value (bar) of the diagonal entries of the MAC matrices computed for all samples and both configurations, comparing the re-simulated model and their respective target response. Again, all mean values are close to 1 (90% with $\overline{\text{MAC}} \geq 0.995$), so an overall excellent updating performance can be stated. Single predictions lead to worse results, as depicted by the minimum value (4.3% of all have a MAC $\leq 0.98$), especially for the higher order modes, though the MAC value of less than 0.8 is only obtained for the 10$^{\text{th}}$ eigenmode of the free-free configuration.

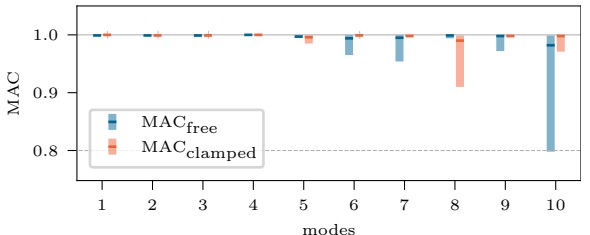

**Figure 17.** Mean, maximum, and minimum diagonal entries of the MAC matrices computed for 1,000 target responses.

The generally good performance is also confirmed by the predicted corresponding natural frequencies. Figure 18 shows the relative error from the re-simulated frequencies to the target frequencies of each mode for both configurations, giving the mean and standard deviation over all re-simulated samples. The range of the mean error is $|\bar{e}_f| < 0.25\%$ and

the standard deviation $\sigma_{e_f} < 1.50\%$.

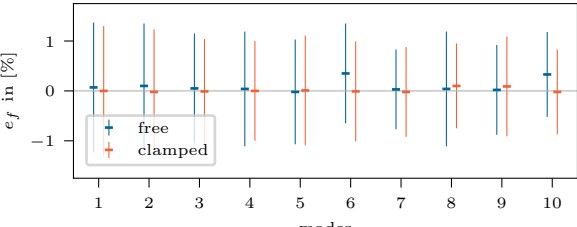

**Figure 18.** Mean and standard deviation of the natural frequency error $e_f$ computed for 1,000 target responses.

The results of the presented resimulation analysis show that:

1. The counteracting intrinsic model ambiguities discussed in Sect. 4.2 cancel each other out, i.e., the overall shell laminate properties are correctly predicted, although the individual stiffness or density of the layers (Biax90 and Triax) are not predicted accurately. So the cINN still correctly captures the global model behavior with respect to mass and stiffness distribution.
2. As expected, the insensitive and thereby excluded input parameters really do not have an impact on the results and can be chosen arbitrarily (cf. Fig. 15).
3. The overall cINN updating performance is strikingly good, with on average 90% of the mode shapes of the resimulated samples showing a $\overline{\text{MAC}} \geq 0.995$. The frequencies were recovered with a mean error of $|\bar{e}_f| < 0.25\%$.

### 4.5 Replacing Sensitivity Analysis

Similar to other model updating studies such as Luczak et al. (2014), this work relies on a sensitivity study to reduce the parameter space of the updating problem to significant parameters. This so-called feature selection is performed in this particular investigation with the aforementioned Sobol method. A quasi-random sampling with low-discrepancy sequences (Dick and Pillichshammer, 2010) is applied to compute the Sobol indicies, which is a computational and space-efficient sampling method for the sensitivity analysis. However, the sampling set to train the cINN in general has to span a real random sampling space, where all features are varied independently, but simultaneously. That means, despite the 79,360 samples for the sensitivity analysis, an additional set of 30,000 samples has to be generated for training purposes and a second variably-sized set for validation and testing of the cINN. In total, this results in approximately 115,000 samples and thus model evaluations. This is crucial considering that the model evaluation in general is the computational bottleneck. Although a classical optimization algorithm would also need a feature selection to reduce the updating problem

complexity on top of its usual model evaluation number for the optimization process, the overhead of the sensitivity cuts down the computational benefit of the cINN. A single model evaluation from creating the input parameter set to importing
the modal response of the model took on average approx. 80 s on a single-core device. We generated the 115,000 samples on a 40-core computing cluster in slightly less than 2.66 days. In contrast, the cINN training for 150 epochs took only 0.67 h on an NVIDIA Tesla P100 GPU.
To reduce the computational sampling time, the idea is to apply the cINN on the full input parameter set $x$ to identify relevant parameters. The cINN implicitly detects irrelevant features by predicting an uncertain posterior distribution, i.e., high standard deviation, due to missing information for the
inference in the response. However, the current Sect. 4 and 4.2 showed that intrinsic model ambiguities lead to wider distributions, without being inaccurate in the global model behavior. This means the respective input parameters should not be rejected due only to a widely distributed posterior pre-
diction. Therefore, we combine three metrics to perform the feature selection on the posterior predictions of the full input parameter set with respect to standardized values:

1. Root mean square error (RMSE) of the predicted posterior's mean and target value
2. Standard deviation of the predicted posterior distribution
3. Cross-correlation matrix of the predicted posterior's mean values

   The RMSE should reject features that might have a nar-
30 row predicted posterior distribution, but do not match the target value. This is more a security or backup metric. The standard deviation is a metric for the confidence of the cINN and should reject features that are not significantly included in the information of the modal beam response. And finally,
a cross-correlation matrix should reveal intrinsic model ambiguities from feature interactions, in order to keep the respective features, though the other two metrics would reject them. The cross correlation matrix of this inverse problem is depicted in Fig. 19. The input $\text{feat}_{40-54}$ and $\text{feat}_{60-74}$
in the matrix correspond to $\rho_{\text{Biax90,N0-N4}}$, $E_{11,\text{Biax90,N0-N4}}$, $G_{12,\text{Biax90,N0-N4}}$ and $\rho_{\text{Triax,N0-N4}}$, $E_{11,\text{Triax,N0-N4}}$, $G_{12,\text{Triax,N0-N4}}$, respectively, which show the high negative correlation of the interacting features discussed in Sect. 4.2. This matrix also helps to detect other relevant correlations. Especially nearby
nodes of the same feature (e.g., $\text{feat}_{85-87}$, $E_{11,\text{Flange,N0-2}}$) can counteract each other, as these have to predict in combination the spline behavior in between them, i.e., if one increases, the other has to diminish. Similar behavior was already detected in Bruns et al. (2019).
Similar to the Sobol threshold $S_{ij,thld} = 0.1$, thresholds for the given metrics can be chosen arbitrarily again and rely on experience. In this case we have chosen $\text{RMSE}_{thld} = 0.5$, $\sigma_{thld} = 0.5$, $\text{XCorr}_{thld,max} = -0.75$. Table A1 lists all

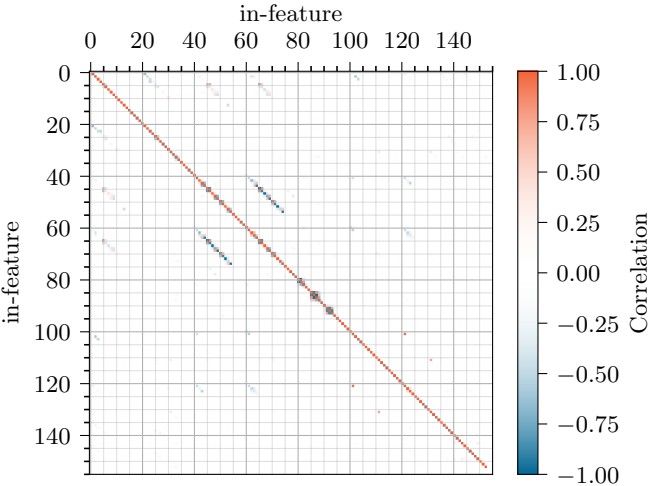

**Figure 19.** Cross-correlation of all input feature based on mean posterior prediction of the 5,000 test samples.

features selected by the sensitivity analysis and the cINN in comparison. The sensitivity analysis selects 49 features, 55 while the cINN includes 54 features. Most of the features agree for both selection methods, except those included in Table 6. The cINN, for example, includes the input features: $E_{11,\text{UD,N4}}$, $G_{12,\text{Balsa,N1}}$, which can be very well predicted by the cINN, but which are not detected by the sensitivity anal- 60 ysis to be significant for the response variations. Additionally, it detects a few highly negative correlating features: $E_{11,\text{Biax90,N4}}$ and $G_{12,\text{Biax90,N0-2,4}}$, which follow the similarly ambiguous behavior shown in the Sect. 4.2, counteracting the respective Triax properties. However, the features: $\rho_{\text{Triax,N1,2}}$, 65 $\rho_{\text{Foam,N1}}$, detected by the sensitivity analysis were excluded by the cINN, though at least the first two show a significant $S_{max} > 0.200$.

**Table 6.** Feature selection discrepancies between both methods: sensitivity analysis (SA) and the cINN-based approach, and their corresponding metrics. All values depicted in bold meet their corresponding threshold and are thus selected by the respective approach.

| Feature | SA | $S_{max}$ | SVD | cINN | RMSE | $\sigma$ | $\text{XCorr}_{min}$ |
|---|---|---|---|---|---|---|---|
| $E_{11,\text{UD,N4}}$ | | 0.006 | | ✓ | **0.340** | **0.354** | -0.4407 |
| $E_{11,\text{Biax90,N4}}$ | | 0.051 | | ✓ | 0.913 | 0.881 | **-0.9524** |
| $G_{12,\text{Biax90,N0}}$ | | 0.040 | | ✓ | 0.862 | 0.833 | **-0.8341** |
| $G_{12,\text{Biax90,N1}}$ | | 0.062 | | ✓ | **0.454** | **0.374** | **-0.8889** |
| $G_{12,\text{Biax90,N2}}$ | | 0.078 | | ✓ | 0.941 | 0.920 | **-0.986** |
| $G_{12,\text{Biax90,N4}}$ | | 0.009 | | ✓ | 1.014 | 0.991 | **-0.9485** |
| $\rho_{\text{Triax,N1}}$ | ✓ | **0.292** | ✓ | | 0.648 | 0.531 | -0.5367 |
| $\rho_{\text{Triax,N2}}$ | ✓ | **0.211** | ✓ | | 0.652 | 0.604 | -0.6785 |
| $G_{\text{Balse,N1}}$ | | 0.017 | | ✓ | **0.285** | **0.230** | -0.2985 |
| $\rho_{\text{Foam,N2}}$ | ✓ | **0.163** | ✓ | | 0.623 | 0.538 | -0.4732 |
| $\rho_{\text{Foam,N3}}$ | | 0.072 | | ✓ | **0.478** | **0.483** | -0.5273 |

Finally, this procedure is based on 30,000 samples and the same cINN architecture and hyperparameters. Figure A2 shows the correlation results for all features included in the sensitivity analysis, where the orange scatter represents the prediction with the model trained on the full input set and the blue scatter the prediction by the former model based on the feature selection from the sensitivity analysis. Only very few features show a significant loss in accuracy compared to the original model, and most likely for the feature with a worse prediction quality. Thus, there is no need to perform a second training process with a reduced data set for the sensitivity-free procedure, though the selection of the samples should still reveal the significant parameters of the model. Relying on the same computing resources mentioned above, the overall process in this particular case adds up to a complete computation time of approximately 20 h, which corresponds to a reduction of 69%. It also reveals that the cINN can handle a higher number of parameters, while extracting the relevant information from the response to predict the significant input features. On account of that, there is no need for a pre-analysing sensitivity study in future investigations. This gives cINN a huge advantage over common approaches as discussed in the introduction. The rely on a sensitivity analysis to identify a significant subspace to reduce the problem dimension. With 30,000 model evaluations for a total of 49 updated features, the cINN is quite efficient. A stochastic updating approach demanded 1,200-12,000 evaluations for a simple 3-feature updating problem (Augustyn et al., 2020; Marwala et al., 2016). Higher dimensional problems could explode in computational costs for common deterministic approaches, even more relying on an additional pre-processed subspace selection (here: 79,000 model evaluations). However, to the best of the authors knowledge, no model updating was found in literature for such a high parameter space as it is presented in this work.

## 5 Conclusions

The current study aims to extend the feasibility study of model updating with *invertible neural networks* presented in Noever-Castelos et al. (2021a) to a more complex and application-oriented level in form of a Tymoshenko beam. The model updating was performed on a global level. This took into account 5-noded splines for input feature representation over the blade span of material density and stiffness, as well as layup geometry. The blade response used for the updating process is in form of modal shapes and frequencies. The outstanding updating results presented in this study strengthens the conclusion in Noever-Castelos et al. (2021a) that *invertible neural networks* are highly capable in efficiently dealing with wind turbine blade model updating for the given global fidelity level.

In comparison with Noever-Castelos et al. (2021a), this investigation increased the model complexity from a single cross-sectional representation to a finite element Tymoshenko beam model of the complete blade. The update parameter space was only slightly expanded for the materials to cover the most relevant, independent elastic properties of orthotropic materials. These, however, are varied over the complete blade length with 3 to 5 noded splines. Moreover, an established, global, variance-based sensitivity analysis with the Sobol method was performed to determine the relevant update parameters. A total of 49 input parameters were updated based on modal responses of the blade in a free-free boundary configuration and a root clamped configuration. The applied cINN approximately doubled its depth and an additional feedforward network was implemented to preprocess the conditions of the cINN in order to improve the network's flexibility and accuracy.

The result analysis of the predicted parameters shows strikingly high coherence with the target values with $R^2$ scores over 0.9 for 75% of the updated parameters. The very high updating certainty of the network is reflected in the narrow predicted posterior distributions of the updated parameters. Moreover, this study revealed more intrinsic model ambiguities of material properties ($E_{11}$, $G_{12}$, $\rho$) of the laminate face sheets Biax90° and Triax due to their proximity in the layup. The cINN learns and understands the intrinsic collinearities of the physical model, which result in ambiguous inverse paths. However, the cINN is still not able to distinguish from which parameter the individual contribution comes. Nevertheless, in contrast to a deterministic approach, the user can see how uncertain the cINN is about the prediction due to its wide spreading of affected feature's prediction. In future contributions this can be handled by updating a joint density or stiffness variation, instead of individual features. However, the resimulation analysis revealed the modal response of the updated models matches the target results exceptionally well, 90% of the mode shapes of the resimulated samples show a $\overline{\text{MAC}} \geq 0.995$ and a mean error in the natural frequencies of $|\bar{e}_f| < 0.25\%$ over 1,000 randomly chosen test samples. Finally, this study presents a method for avoiding the computationally expensive sensitivity analysis by fully exploiting the opportunities of the cINN. For this reason, the full parameter set of $D_{tot} = 153$ was used for the update process. Thanks to the underlying probabilistic approach of the cINN, a similar set of significant input features was detected from the complete parameter space, based on the predicted posterior distributions and a cross correlation between the input feature to identify the ambiguities. Thus, the necessary sample number for the complete process was reduced to 30,000 samples and the computational time by 69%, while maintaining similar outstanding updating results.

Referring back to the three major problems of the approaches studied in the introduction, the cINN tackles these by:

1. A high computational efficiency in relation to the model complexity, i.e., updating parameter space. Even more by the evading computationally expensive sensitivity analysis. The cINN only demanded 30,000 model evaluations ($\approx$20 h) for a total of 49 updated features within an original space of 153 features.
2. An inherent probabilistic evaluation, as it follows the Bayes' theorem and is trained to minimize the negative log-likelihood of the mapping between posterior distribution and latent distribution.
3. Representing a surrogate of the inverted model. By that, the cINN can be evaluated for any given response (in the model boundaries) at practically not additional costs after training. Any other approach is solved only for one particular model response and has to be repeated in case of a different set of response.

In conclusion, the feasibility study was highly successfully extended to a full blade beam model, though with a still limited parameter set. The cINN proved to be extremely capable of performing an efficient model updating with a larger parameter space. The physical model complexity in form of a Tymoshenko finite element beam is already at the state of the art level applied in industry. However, to ensure that the cINN learns the complete inverted physical model, it is important that all possibly relevant parameters have to be varied, so that the cINN is trained for all circumstances of variations for the model updating. Therefore, ongoing and future investigations should bring this method to a real life application, where the parameter space will span more relevant aspects of blade manufacturing deviations, such as e.g., adhesive joints. Moreover, the combined laminate properties of the face sheets might be able to prevent the model ambiguities and even to improve the already good prediction accuracy. One possible application scenario could be a final quality control after manufacturing, if the response generation can be automated. The benefit would be to find rough manufacturing deviations and even provide individually updated models for each blade, which could for example enhance turbine controls.

**Code and data availability.** Code and data available in a publicly accessible repository:
https://github.com/IWES-LUH/Beam-ModelUpdating-cINN

## Appendix A: Tables & Figures

**Table A1.** Comparison of the feature selection performed by the sensitivity analysis (SA) and directly with the cINN applied to the full input parameter set.

| Feature | $S_{\max i}$ | SVD | cINN | Feature | $S_{\max i}$ | SVD | cINN |
|---|---|---|---|---|---|---|---|
| $\rho_{\text{UD,N0}}$ | ✓ | ✓ | ✓ | $E_{11,\text{Triax,N1}}$ | ✓ | ✓ | ✓ |
| $\rho_{\text{UD,N2}}$ | ✓ | ✓ | ✓ | $E_{11,\text{Triax,N2}}$ | ✓ | ✓ | ✓ |
| $\rho_{\text{UD,N3}}$ | ✓ | ✓ | ✓ | $E_{11,\text{Triax,N3}}$ | ✓ | ✓ | ✓ |
| $E_{11,\text{UD,N0}}$ | ✓ | ✓ | ✓ | $E_{11,\text{Triax,N4}}$ | ✓ | ✓ | ✓ |
| $E_{11,\text{UD,N1}}$ | ✓ | ✓ | ✓ | $G_{12,\text{Triax,N0}}$ | ✓ | ✓ | ✓ |
| $E_{11,\text{UD,N2}}$ | ✓ | ✓ | ✓ | $G_{12,\text{Triax,N1}}$ | ✓ | ✓ | ✓ |
| $E_{11,\text{UD,N3}}$ | ✓ | ✓ | ✓ | $G_{12,\text{Triax,N2}}$ | ✓ | ✓ | ✓ |
| $E_{11,\text{UD,N4}}$ | | | ✓ | $G_{12,\text{Triax,N3}}$ | ✓ | ✓ | ✓ |
| $G_{12,\text{Biax45,N0}}$ | | ✓ | ✓ | $G_{12,\text{Triax,N4}}$ | ✓ | ✓ | ✓ |
| $G_{12,\text{Biax45,N1}}$ | ✓ | ✓ | ✓ | $\rho_{\text{Flange,N0}}$ | ✓ | | ✓ |
| $G_{12,\text{Biax45,N2}}$ | ✓ | ✓ | ✓ | $\rho_{\text{Flange,N1}}$ | ✓ | ✓ | ✓ |
| $G_{12,\text{Biax45,N3}}$ | ✓ | ✓ | ✓ | $E_{11,\text{Flange,N0}}$ | | ✓ | ✓ |
| $\rho_{\text{Biax9,N3}}$ | ✓ | ✓ | ✓ | $E_{11,\text{Flange,N1}}$ | ✓ | ✓ | ✓ |
| $\rho_{\text{Biax9,N4}}$ | ✓ | | ✓ | $E_{11,\text{Flange,N2}}$ | ✓ | ✓ | ✓ |
| $E_{11,\text{Biax9,N0}}$ | ✓ | ✓ | ✓ | $E_{11,\text{Flange,N3}}$ | | ✓ | ✓ |
| $E_{11,\text{Biax9,N1}}$ | ✓ | | ✓ | $G_{12,\text{Flange,N1}}$ | ✓ | ✓ | ✓ |
| $E_{11,\text{Biax9,N2}}$ | ✓ | ✓ | ✓ | $G_{12,\text{Flange,N2}}$ | ✓ | ✓ | ✓ |
| $E_{11,\text{Biax9,N3}}$ | ✓ | ✓ | ✓ | $G_{12,\text{Flange,N3}}$ | | ✓ | ✓ |
| $E_{11,\text{Biax9,N4}}$ | | | ✓ | $\rho_{\text{Balsa,N1}}$ | ✓ | ✓ | ✓ |
| $G_{12,\text{Biax9,N0}}$ | | | ✓ | $G_{\text{Balse,N1}}$ | | | ✓ |
| $G_{12,\text{Biax9,N1}}$ | | | ✓ | $\rho_{\text{Foam,N2}}$ | ✓ | ✓ | |
| $G_{12,\text{Biax9,N2}}$ | | | ✓ | $\rho_{\text{Foam,N3}}$ | | | ✓ |
| $G_{12,\text{Biax9,N3}}$ | ✓ | ✓ | ✓ | $P_{\text{SS,Mid,spar cap,N0}}$ | ✓ | ✓ | ✓ |
| $G_{12,\text{Biax9,N4}}$ | | | ✓ | $P_{\text{SS,Mid,spar cap,N1}}$ | ✓ | ✓ | ✓ |
| $\rho_{\text{Triax,N1}}$ | ✓ | ✓ | | $P_{\text{SS,Mid,spar cap,N2}}$ | ✓ | ✓ | ✓ |
| $\rho_{\text{Triax,N2}}$ | ✓ | ✓ | | $P_{\text{PS,Mid,spar cap,N0}}$ | ✓ | ✓ | ✓ |
| $\rho_{\text{Triax,N3}}$ | ✓ | ✓ | ✓ | $P_{\text{PS,Mid,spar cap,N1}}$ | ✓ | ✓ | ✓ |
| $\rho_{\text{Triax,N4}}$ | ✓ | ✓ | ✓ | $P_{\text{PS,Mid,spar cap,N2}}$ | ✓ | ✓ | ✓ |
| $E_{11,\text{Triax,N0}}$ | ✓ | ✓ | ✓ | | | | |

**Table A2.** Identified mode shapes of the first 10 modes (excluding rigid body motion) of the free-free and the clamped modal configuration.

| Mode no. | Free-free | Clamped |
|---|---|---|
| 1 | 1.Flap | 1. Flap |
| 2 | 1. Edge | 1. Edge |
| 3 | 2. Flap | 2. Flap |
| 4 | 1.Torsion | 2. Edge |
| 5 | 3. Flap | 3. Flap |
| 6 | 2. Edge | 1. Torsion |
| 7 | 4. Flap | 4. Flap |
| 8 | 2. Torsion | 2. Torsion |
| 9 | 5. Flap | 3. Torsion |
| 10 | 3. Edge | 5. Flap |

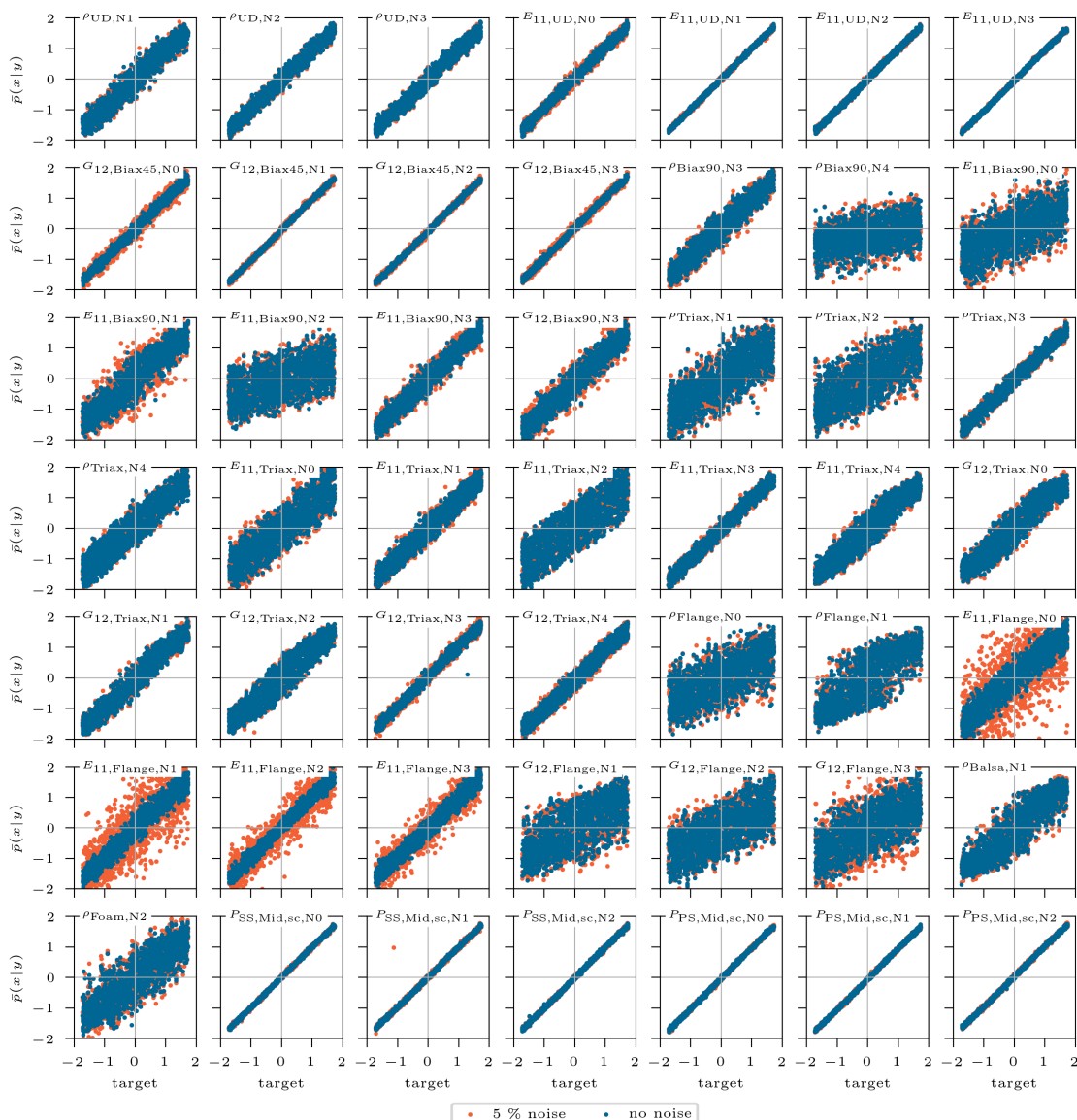

**Figure A1.** Standardized mean of posterior prediction $\bar{x}$ of the updated inputs over the corresponding target standardized value for the 5,000 test samples. The original samples predicted with clean conditions in blue, compared to samples with noisy flawed conditions (5% random noise) in orange. The noisy conditions are intended to simulate measurement inaccuracies of the modal beam response.

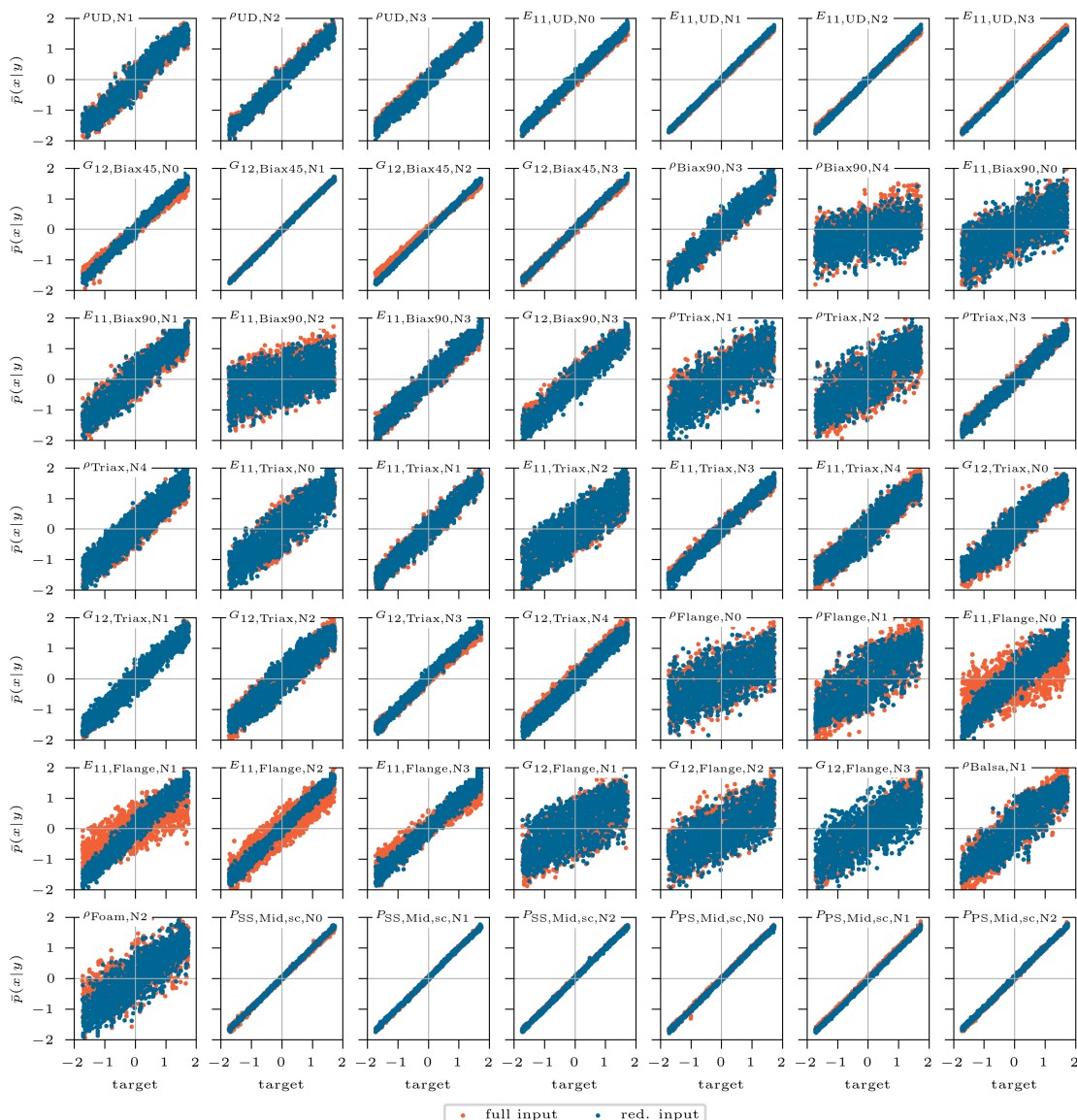

**Figure A2.** Standardized mean of posterior prediction $\bar{x}$ of the inputs selected by the sensitivity analysis, over the corresponding target standardized values for the 5,000 test samples. The original samples predicted with the reduced input set according to the sensitivity analysis selection are depicted in blue. They are compared with the inputs predicted by the cINN trained on the full input set (in orange).

**Author contributions.** P.N-C.: conceptualization, methodology, software, validation, formal analysis, investigation, resources, data curation, writing–original draft preparation, writing–review and editing, visualization; D.M.: software, investigation, writing–review and editing; C.B.: writing–review and editing, supervision, project administration, funding acquisition. All authors have read and agreed to the published version of the manuscript.

**Competing interests.** The authors declare that they do not have any conflicts of interests.

**Disclaimer.** The information in this paper is provided as is and no guarantee or warranty is given that the information is fit for any particular purpose. The user thereof uses the information at its own risk and liability.

**Acknowledgements.** This work was supported by the Federal Ministry for Economic Affairs and Energy of Germany (BMWi) in the project ReliaBlade (grant numbers 0324335A/B).

This work was supported by the compute cluster, which is funded by the Leibniz University Hannover, the Lower Saxony Ministry of Science and Culture (MWK), and the German Research Association (DFG).

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
