# Peer review of "Model updating of a wind turbine blade finite element Timoshenko beam model with invertible neural networks"

_Wind Energy Science, 2021_

## Referee Comment (RC1)

Review of the paper entitled "Model updating of a wind turbine blade finite element beam model with invertible neural networks", by Pablo Noever-Castelos et al.

General Comment

The paper deals with an updating procedure for complex wind turbine blade models. The methodology uses modal data (frequencies and shapes) to estimate the properties of the model (Young's and shear moduli, densities, etc…) through invertible neural networks. It seems that the work is totally built on a technique developed by the Author in a previous publication (Pablo Noever-Castelos, *Model updating of wind turbine blade cross sections with invertible neural networks*, submitted to Wind Energy). The paper is well-written and clear enough (I believe that a researcher unfamiliar with neural network should be able to get the main points of the work). The presented results generally support the goodness of the proposed methodology, even if they may be better commented.

There are three main concerns that should be addressed to provide a paper which is worth publishing, related to 1) link between present and previous publication of the Authors, 2) the collinearity (ambiguities) among features and 3) interpretation of the obtained results in light of the Sobol indices. These three concerns are better explained in "Major comments". This document also lists some "minor comments", that I hope the Author will accommodate in a revised version of the manuscript.

My final recommendation is to accept the manuscript only if the three "Major comments" will be adequately addressed.

Major comments

1. The link between the previous paper of the same Authors (*Model updating of wind turbine blade cross sections with invertible neural networks*) and the present one should be better highlighted, and the differences clearly stated. As far as I have understood, in the previous paper, only the final sectional properties are considered, while here the blade model is more complex and comprises detailed sectional descriptions. Moreover, the method for replacing the sensitivity computations (see sec. 4.5) is new. If so, the Authors should also comment on the need of such an addition of complexity. What is the eventual balance between a higher complexity in the model (and in the updating process) and the performance potentially achievable? Does it worth it? Finally, probably the word "beam" in the title indicates a simpler blade model with respect to that used in the work which is characterized by a fully three-dimensional description. Please, check.

2. In Section 2.3 "Selection of Sobol Indices", the Authors use the Sobol Indices to select only those parameters which significantly affect the system response (see lines 150-151 "we aim to consider only features which have a significant impact during at least one event at one location, thus containing enough information for the updating process."). This is totally correct, but it is certainly possible that combinations of input features may lead to similar outputs. This implies that one cannot comprehend 'who does what', and in turn cannot generate a robust model. During the identification of physics-based models, this problem is called collinearity (ambiguity, in this paper) and is handled by looking at the sensitivity matrix (e.g., SVD of sensitivity matrix, Cramèr-Rao bounds). Even if neural networks are employed, the collinearity problem should stay the same, because it refers to the intrinsic properties of the system. Since the Sobol indices are just metrics to quantify the sensitivity of outputs respect inputs, I imagine that collinearity problem (if present) may be found looking at such indices as well. The Author should comment and possibly extend the treatment.

3. Interpretation of the results in Section 4.1, (see especially fig. 9): At this point, it is essential to create the link between the goodness of the prediction and the Sobol indices. This will ease the comprehension of the results. For example, feature33 has a high Sobol index but an accuracy rather poor (R^2=0.8, with a significant spread around the regression line). On the other side, feature4 has a very low Sobol index (0.11 close to the selected threshold) but has an excellent accuracy. Why? I imagine that the link between the sensitivity analysis and the estimation accuracy should be stronger than what we see in such results. Please, comment and possibly explain thoroughly the obtained results.

Minor comments

- Line 85: Please, correct "resimulationn"
- Line 95: Please, remove comma in "Sobol derived, the 1$^{st}$ order Sobol index…"
- Line 113: symbol "$N_{FE}$" appears here for the first time, but its meaning was not previously defined.
- Line 131: "All applied variances are approximately twice the permitted manufacturing tolerances". Probably with the word "Variances" the Authors refers to the difference imposed to the parameters to perform the sensitivity analysis. If so, the word "variances" could be misleading as it often indicates a statistic metric.
- Line 141: does the sentence "… and the six degrees of freedom of each finite element beam node NFE are saved and…" refer to modal shapes?
- Tab. 2: It would be interesting to plot Sobol indices as function of the blade span, parameterized with respect to the typology of the element, so as to give an idea on how the observability changes as function of the blade span. The Author may try convert the table into a plot.
- Section 3: the description of the network could be improved. In particular, it could be important
  o To clarify what is new with respect to previous works; it seems that the network is totally built on previous activities, and no dedicated updates were conducted for the present research.
  o To clarify the reason why this network type is better suited to the application at hand. From this point of view, I would expect here a connection with the Introduction, and especially with the three points listed in Sec. 1.2. Why is the present network able to handle more complex problem than those already studied in literature? How can the present network evaluate uncertainty in the results? Why is the present network able to create a generalize model not focused on a particular condition?
- Line 231: check spelling of "resimulationn".
- Section 4.1: The analysis is good and interesting, but here comes again the main question: as in a major comment, is it possible that the poor accuracy of some parameters may be connected to collinearity problems?
- Section 4.1, fig. 9: Instead of using in "Feat_x", the reader could benefit from subtitles with the physical meaning. So, he/she does not have to jump to Tab.2
- Section 4.1, fig. 9: At this point, it is essential to create the link between the goodness of the prediction and the Sobol indices. This will ease the comprehension of the results. For example, feature33 has a high Sobol index but an accuracy rather poor (R^2=0.8, with a significant spread around the regression line). On the other side, feature4 has a very low Sobol index (0.11 close to the selected threshold) but has an excellent accuracy. Why? I imagine that the link between the sensitivity analysis and the estimation accuracy should be stronger than what we see in such results.

- Lines 316-322: These lines and the previous section talk about something that I had in mind since the beginning of the manuscript: the different properties of each section (Young modulus, densities, etc....) may contribute together to the final sectional stiffness, and eventually it is hard to distinguish among those properties looking at global pieces of information (modal data). This, however, refers to an intrinsic problem of the systems. When Authors write "we can state that the cINN should correctly predict the total mass and the stiffness contributions in a global manner ...", at least for me, they report something rather obvious. Please, comment and, if needed, clarify.
- Fig. 15: The analysis underlying this plot is interesting. I was wandering whether a similar conclusion can be derived from the Sobol analysis of Sec. 2.1. I guess that features belonging to blade root and tip be associated to both lower estimation accuracy (see Fig 15) and lower Sobol indices (from Tab. 2). Please, verify and comment.
- Line 386: "cINN correctly captures the global model behavior with respect to mass and stiffness distribution.". What about the blade center of gravity position, which is a value simple to be measured? This data can be used in the estimation process. Was it done?
- Section conclusion:
  - The sentence "invertible neural networks are highly capable to efficiently dealing even with an extensive wind turbine blade model updating" should be better explained. In fact, the estimation problem is solved but still the updating process results accurate only for global model characteristics (see line 386: "the cINN correctly captures the global model behavior with respect to mass and stiffness distribution"). I suggest stressing this fact.
  - Lines 461-462: "The ambiguities are captured very accurately by the network.". What do the Authors mean with this sentence? Does it mean that the cINN is able to get rid of ambiguities and not-identifiable combinations of features and perform the estimation accurately for the rest of the features? If so, maybe the sentence should be clarified.

---

## Author Response (AR2)

*Dear Reviewer,*

*Thank you very much for taking the time to study this paper and provide valuable constructive criticism, which we believe has helped to develop and strengthen this work significantly. Please find our answers to all your comments, by either corrected or added text sections or comments on your suggestions/concerns. We hope to have correctly understood and adequately accommodated all your concerns. Thank you again for your contribution.*

Major comments

1. The link between the previous paper of the same Authors (*Model updating of wind turbine blade cross sections with invertible neural networks*) and the present one should be better highlighted, and the differences clearly stated. As far as we have understood, in the previous paper, only the final sectional properties are considered, while here the blade model is more complex and comprises detailed sectional descriptions. Moreover, the method for replacing the sensitivity computations (see sec. 4.5) is new.

   *Thank you for this comment, we fully agree that the manuscript lacks a clear demarcation of both studies. We have adapted Sec. 1.3 to highlight the differences to the previous study:*

   *"The specific objective of this current investigation in contrast to the previous publication (Noever-Castelos et al., 2021a) is to:*
   1. *Extend the feasibility study and methodology to a complete three-dimensional finite element Tymoshenko beam model of a wind turbine blade, instead of analyzing isolated cross-sections*
   2. *Introduce parameter splines for the input variation along the blade*
   3. *Use modal blade shapes and frequencies as model response*
   4. *Replace the sensitivity analysis for the parameter subspace selection by the global variance-based Sobol method (Sobol', 1993), which takes interactions of input parameters into account*
   5. *Implement a pre-processing feed-forward neural network for the cINN conditions*
   6. *Analyze the potential of replacing or neglecting the sensitivity analysis by training the cINN on the full parameter space"*

   *We hope this sufficiently differentiates the two studies.*

   If so, the Authors should also comment on the need of such an addition of complexity. What is the eventual balance between a higher complexity in the model (and in the updating process) and the performance potentially achievable? Does it worth it?

*We do not fully understand the reference of this question.*

    a) *If you mean the replaced sensitivity computation; this reduces dramatically the computational costs. See Sec. 4.5:*

        *"That means, despite the 79,360 samples for the sensitivity analysis, an additional set of 30,000 samples has to be generated for training purposes and a second variably-sized set for validation and testing of the cINN."*

        *The method with no sensitivity analysis neglects the 79,360 samples for the sensitivity and thus:*

        *"Relying on the same computing resources mentioned above, the overall process in this particular case adds up to a complete computation time of approximately 20 h, which corresponds to a reduction of 69%."*

    b) *If you relate to the increased model complexity; We are now switching from an isolated single cross-sectional perspective to a complete 3D Tymoshenko beam, built up from several (>50) cross-sections. By that, we have stepped to a model complexity applied in real world problems. Therefore, it is necessary to reach an applicable model level. We hope this becomes clear in the new listing above.*

**Finally, probably the word "beam" in the title indicates a simpler blade model with respect to that used in the work which is characterized by a fully three-dimensional description. Please, check.**

*This is a good note. We have added Tymoshenko in the title and named it at different locations in the text to make it clear.*

2. In Section 2.3 "Selection of Sobol Indices", the Authors use the Sobol Indices to select only those parameters which significantly affect the system response (see lines 150-151 "we aim to consider only features which have a significant impact during at least one event at one location, thus containing enough information for the updating process.". This is totally correct, but it is certainly possible that combinations of input features may lead to similar outputs. This implies that one cannot comprehend 'who does what', and in turn cannot generate a robust model. During the identification of physics-based models, this problem is called collinearity (ambiguity, in this paper) and is handled by looking at the sensitivity matrix (e.g., SVD of sensitivity matrix, Cramèr-Rao bounds). Even if neural networks are employed, the collinearity problem should stay the same, because it refers to the intrinsic properties of the system. Since the Sobol indices are just metrics to quantify the sensitivity of outputs respect inputs, I imagine that collinearity problem (if present) may be found looking at such indices as well. The Author should comment and possibly extend the treatment.

*We fully agree that it is a restricted view on the phyiscal model, if only the first order Sobol index is examined. And that intrinsic collinearities of parameter influences may be and actually are implied in the physical model is also correct. Now to fully understand the physical model it is of course interesting and important to analyze $2^{nd}$ and probably $3^{rd}$ order Sobol indices to extract any interactions. However, the purpose of this sensitivity*

*analysis is to have an easy preselection of a parameter subspace to analyze. Now there are two ways of achieving this, either by computing the $1^{st}$ and $2^{nd}$ order Sobol index, which increases the computational cost by 2 or by analyzing the $1^{st}$ and total order Sobol index, which can be calculated at the same cost as the $1^{st}$ order index. The total order Sobol index includes the $1^{st}$ and all higher order Sobol indicies. If the model is not purely additive, i.e., interactions of the model parameters exist, then the total index exceeds one for the analyzed response. By that, an existence of collinearity is proofed, however it is not shown which parameters are collinear. However, this is not that essential for the sake of selecting an appropriate parameter subspace.*

*Concerning the method how to evaluate the indices, it is probably relevant to include not only the parameters, which exceed the threshold for the max. Sobol Index, but also those, that have on average a high contribution. You have suggested using the singular value decomposition SVD; actually we have found a publication dealing with subspace selection from a sensitivity analysis, that uses the SVD to reduce the dimensionality of the sensitivity matrix. After reducing the resulting singular value matrix to the p most contributing dimensions, it is followed by a QR factorization to map this again onto the real parameter space. Similar to principal component analysis (PCA) this reduces the parameter space to the most contributing parameters, where the contribution is measured with the total Sobol index.*

*Finally, we have chosen a combination, we did the max. Sobol Index threshold and the SVD-QR method and combined both selected subspaces to the final parameter subspace. As far as we can judge, this should include all your pronounced concerns on the subspace selection method.*

*And you are totally right, that the cINN includes all this intrinsic collinearities of the physical model. And this is, what we can see in the further analysis in Section 4.2 "Intrinsic Ambiguities" and 4.5 the Cross-Correlation. However, concerning the computational cost, it is much more expensive to calculate the $2^{nd}$ order Sobol Index than to analyze the cINN predictions with a cross correlation matrix. And as model updating approaches are computationally very expensive, we have settled with the indepth analysis of the cINN predictions rather than the second order Sobol index. We hope this makes sense to you, though the subspace selection should now also consider interacting parameters.*

*We will refrain from posting every particular change in the comment here, but all we have addressed above is included in the adaptions of Section 2.*

*The wording ambiguity for collinearity, originates from the original publications to INN (Analyzing Inverse Problems with Invertible Neural Networks, Ardizzone et. al., 2019) and was also used in the first feasibility study and is kept here for consistency.*

3. Interpretation of the results in Section 4.1, (see especially fig. 9): At this point, it is essential to create the link between the goodness of the prediction and the Sobol indices. This will ease the comprehension of the results. For example, feature33 ($\rho_{Flange,N1}$) has a high Sobol index but an accuracy rather poor ($R^2=0.8$, with a significant spread around the regression line). On the other side, feature4 ($E_{11,UD,N0}$) has a very low Sobol index (0.11 close to the selected threshold) but has an excellent accuracy. Why? I imagine that the link

between the sensitivity analysis and the estimation accuracy should be stronger than what we see in such results. Please, comment and possibly explain thoroughly the obtained results.

*Thank you for that notice, you are right, the link between the Sobol indices and the prediction accuracies was missing. We have added a paragraph to the end of the Section 4.1 with a table highlighting and explaining the most strinking discrepencies of sensitivity index to prediction accuracy. In short the intrinsic collinearity of the physical model is the key problem, why high Sobol indices yet yield in low prediction accuracy. In the forward path (physical model), collinearities are easy to track as they add up to the final response, though the inverse path (cINN) has to map ambiguous responses to a set of possible input features. However, if this ambiguity is not existence, i.e., an input feature has no substantial collinearity with other features, then even a low sensitivity may be sufficient to map an input feature to only a very few determined responses. Please find a more detailed description at the end of Section 4.1. We hope this satisfies your concern.*

Minor comments
- Line 85: Please, correct "resimulationn"
  *Corrected here and at different locations in the text. Thank you for that comment.*
- Line 95: Please, remove comma in "Sobol derived, the 1st order Sobol index..."
  *Corrected.*
- Line 113: symbol "$N_{FE}$" appears here for the first time, but its meaning was not previously defined.
  *The sentence was rephrased to:*
  *"Thus, the finite element model consists of 51 nodes ($N_{FE}$)"*
- Line 131: "All applied variances are approximately twice the permitted manufacturing tolerances". Probably with the word "Variances" the Authors refers to the difference imposed to the parameters to perform the sensitivity analysis. If so, the word "variances" could be misleading as it often indicates a statistic metric.
  *Thank you. This should of course mean "variations".*
- Line 141: does the sentence "... and the six degrees of freedom of each finite element beam node NFE are saved and..." refer to modal shapes?
  *The sentence was rephrased to:*
  *"For all 10 mode shapes of each configuration (free-free and clamped), the natural frequency and the three deflections and three rotations of each finite element beam node $N_{FE}$ are saved."*
- Tab. 2: It would be interesting to plot Sobol indices as function of the blade span, parameterized with respect to the typology of the element, so as to give an idea on how the observability changes as function of the blade span. The Author may try convert the table into a plot.
  *Although we have already suggested in a previous comment in the interactive discussion to plot the sensitivity matrix, which connects to the stated suggestion here, we have tried several plots. However, putting all information into one plot was impossible, while maintaining a good readability and providing an interpretable plot. For several presentations, which have been prepared on that topic, we could only come*

*up with a plot as shown below. But this only represents one input spline, i.e., 5 Nodes, of a total of 33 splines.*

*Just for the understanding of the plot: The figure shows two plots, one for the modes shapes of the free vibration (left) and one for the clamped vibration (right) for the $E_{11,UD}$ parameter. Each plot is divided into the five nodes of the spline along the x-Axis. For each Node the 10 mode shapes of each configuration are plotted over the radius (y-Axis). That means each points represents one finite element node of the beam for the respective mode shape. However, it was still necessary to collapse the dimension of the DOFs, i.e., each point only takes the maximum value of the $1^{st}$ order Sobol index of all 6 DOFs of that finite element node.*

*Therefore, in the end it would have been either too many necessary plots or too much information to condense into one plot. That is why we have to apologize and refrain from trying to visualize the sensitivity matrix. We hope this finds your approval.*

[Figure]

- Section 3: the description of the network could be improved. In particular, it could be important.
    - To clarify what is new with respect to previous works; it seems that the network is totally built on previous activities, and no dedicated updates were conducted for the present research.

        *Yes, it is right, that the network is built on previous activities concerning the cINN. However, this present research includes an update as stated in Section 3 in line 182-189:*

        *"However, unlike the underlying feasibility study Noever-Castelos et al. (2021a), an additional feedforward network is implemented, referred to as a conditional network (violet). The idea is to preprocess the raw conditions c, i.e., beam responses, before passing them to the sub-networks in the CCs. It is trained in*

*conjunction with the cINN, to extract relevant feature information optimally for each stage. The conditional network architecture is inspired by Ardizzone et al. (2019b) and should extract higher-level features of c to feed into the sequential CCs, which, according to Ardizzone et al. (2019b), should relieve the sub-networks from having to relearn these higher-level features each time again. With a conditional beam response c of shape dim(c) = dim($N_{FE,sel}$) x dim(y), the conditional network applies 1D-convolutions (conv 1D) to process the data, which gradually increase in size to progressively extract higher-level features"*

- o To clarify the reason why this network type is better suited to the application at hand. From this point of view, I would expect here a connection with the Introduction, and especially with the three points listed in Sec. 1.2. Why is the present network able to handle more complex problem than those already studied in literature?

  *This was added to the end of section 4.5:*

  *"This gives cINN a huge advantage over common approaches as discussed in the introduction. The rely on a sensitivity analysis to identify a significant subspace to reduce the problem dimension. With 30,000 model evaluations for a total of 49 updated features, the cINN is quite efficient. A stochastic updating approach demanded 1,200-12,000 evaluations for a simple 3-feature updating problem (Augustyn et al., 2020; Marwala et al., 2016). Higher dimensional problems could explode in computational costs for common deterministic approaches, even more relying on an additional pre-processed subspace selection (here: 79,000 model evaluations). However, to the best of the authors knowledge, no model updating was found in literature for such a high parameter space as it is presented in this work."*

- o How can the present network evaluate uncertainty in the results?

  *The method is only scratched on the surface, as it is covered in the previous publication. However, we agree that at least a rough explanation should be provided with a reference to other publications for more in-depth information. Therefore, we have added extended the paragraph at the beginning of Section 4:*
  *"The concept and training of the cINN is based on the Bayes' theorem to infer a posterior distribution $p_x(x|c)$ from a set of conditions c. Therefore, the cINN learns the conditioned transformation from the posterior distribution $p_x(x|c)$ onto the latent distribution $p_z(z)$, as depicted in Fig. 6. This mapping can be achieved through maximum likelihood training. The training is performed over 150 epochs, i.e., training iterations, with a samples size of 30,000 training samples, in order to minimize the negative log-likelihood $L_{NLL}$ (given in Eq. (9)). For a more detailed description of the inherent method of cINNs please refer to Noever-Castelos et al. (2021b); Ardizzone et al. (2019a)."*

- o Why is the present network able to create a generalize model not focused on a particular condition?

  *This is a good comment. We like the idea of closing the circle of naming the problems and providing this solution. Therefore, we have moved this to the conclusion section, where we have integrated the following paragraph:*
  *"Referring back to the three major problems of the approaches studied in the introduction, the cINN tackles these by:*

- *A high computational efficiency in relation to the model complexity, i.e., updating parameter space. Even more by the evading computationally expensive sensitivity analysis. The cINN only demanded 30,000 model evaluations (20h) for a total of 49 features within an original space of 153 features.*
- *An inherent probabilistic evaluation, as it follows the Bayes' theorem and is trained to minimize the negative log-likelihood of the mapping between posterior distribution and latent distribution.*
- *Representing a surrogate of the inverted model. By that, the cINN can be evaluated for any given response (in the model boundaries) at practically not additional costs after training. Any other approach is solved only for one particular model response and has to be repeated in case of a different set of response."*

- Section 4.1: The analysis is good and interesting, but here comes again the main question: as in a major comment, is it possible that the poor accuracy of some parameters may be connected to collinearity problems?
  *This is correct and as the analysis evolves/continues these collinearities/ambiguities are highlighted and connected to the poor accuracy of the prediction.*
- Section 4.1, fig. 9: Instead of using in "Feat_x", the reader could benefit from subtitles with the physical meaning. So, he/she does not have to jump to Tab.2
  *We have adapted the descriptions of the plots, which really improved the readability, thanks.*
- Lines 316-322: These lines and the previous section talk about something that I had in mind since the beginning of the manuscript: the different properties of each section (Young modulus, densities, etc....) may contribute together to the final sectional stiffness, and eventually it is hard to distinguish among those properties looking at global pieces of information (modal data). This, however, refers to an intrinsic problem of the systems. When Authors write "we can state that the cINN should correctly predict the total mass and the stiffness contributions in a global manner ...", at least for me, they report something rather obvious. Please, comment and, if needed, clarify.

  *After talking with the authors of the original publications to cINNs and INNs, they were always interested if the cINN can accurately capture such ambiguities/collinearities in the inverse path. Of course, in the forward path it is obvious what a "surrogate model" does with such collinearities, as they kind of sum up, however, in the inverse path it is not clear, as the "inverse surrogate model" does not know from which parameter the overall contribution comes from. Concerning the overall stiffness of the laminate a classical metaheuristic optimization approach could easily dedicate 100% to the first layer and 0% to the second, while achieving the right result. The cINN, however, learns from the given samples, a reasonable range for each layer, and give some more realistic results and they recognize the correct fraction of each layer contribution to the overall stiffness (see Sec. 4.2). And at the end of the paragraph, we have stated an easy solution for that issue by varying the properties of the laminate as whole and not the different layers. This should exclude this collinearities/ambiguities. we hope this makes clear why it is still interesting for some readers to see this behavior, while it is deductible from physical knowledge.*

- Fig. 15: The analysis underlying this plot is interesting. I was wandering whether a similar conclusion can be derived from the Sobol analysis of Sec. 2.1. I guess that features belonging to blade root and tip be associated to both lower estimation accuracy (see Fig 15) and lower Sobol indices (from Tab. 2). Please, verify and comment.

*Interesting idea. Parting from the point that the root and tip node of the variation splines only contribute to one respective side of the variation spline and the intermediate nodes to both sides, one could expect that these tip/root nodes have less impact on the modal response and thus on the sensitivity matrix. This would imply that the mode shapes include less information for the updating process. So your anticipation is most of the times right, but it does not hold necessarily for all cases, especially for the root nodes. By having information of the clamped and free vibrating configuration, it is possible to recover information on density (free vibration) and stiffness (clamped vibration) in the root section. However, we do not have this configuration for the tip section, thus here especially the density can be better recovered than the stiffness, as the tip has no boundary condition. What in most cases is visible is that the significant sensitivity indices move from root to tip with the node number of the parameter splines. The figure provided for the previous comment, shows clearly:*

   - *Node 1 (0m) can be found in the clamped configuration only contributing in the root section*
   - *Node 2-4 higher impact on their particular position (5m,10m,15m) but also contributions to the rest*
   - *Node 5 (20m) basically no contribution.*

*And comparing to the cINN prediction on the basis of RMSE to the target values, the figure below shows, what you have expected for most of the cases: Root and Tip Node have low Sobol indices and also poor predictions of the cINN.*

[Figure]

*However, as shown in the response to your last major comment, the cINN only needs one DOF of any mode shape, where the considered parameter contributes as one of the top input parameters, to retrieve enough information for the mapping of output to input parameter. Thus explains why in few cases the cINN can still compute a good prediction, although the Sobol index is low.*

*We hope this answers your question or better confirms your anticipation. However, if you agree, we would still leave this analysis out, to not over inflate the presented study.*

- Line 386: "cINN correctly captures the global model behavior with respect to mass and stiffness distribution.". What about the blade center of gravity position, which is a value simple to be measured? This data can be used in the estimation process. Was it done?

  *No, it was not used. But the center of gravity position should be approximated good as long the mode shapes are accurately captured as these include the mass distribution and by that inherently the total mass and center of gravity location.*

- Section conclusion:
  - The sentence "invertible neural networks are highly capable to efficiently dealing even with an extensive wind turbine blade model updating" should be better explained. In fact, the estimation problem is solved but still the updating process results accurate only for global model characteristics (see line 386: "the cINN correctly captures the global model behavior with respect to mass and stiffness distribution"). I suggest stressing this fact.

    *We have adapted this paragraph. We hope this follows your given idea:*

    *"The model updating was performed on a global level. This took into account 5-noded splines for input feature representation over the blade span of material density and stiffness, as well as layup geometry. The blade response used for the updating process is in form of modal shapes and frequencies. The outstanding updating results presented in this study strengthens the conclusion in Noever-Castelos et al. (2021a) that invertible neural networks are highly capable to efficiently dealing with a full wind turbine blade model updating for the given global fidelity level."*

  - Lines 461-462: "The ambiguities are captured very accurately by the network.". What do the Authors mean with this sentence? Does it mean that the cINN is able to get rid of ambiguities and not-identifiable combinations of features and perform the estimation accurately for the rest of the features? If so, maybe the sentence should be clarified.

    *"The cINN learns and understands the intrinsic collinearities of the physical model, which result in ambiguous inverse paths. However, the cINN is still not able to distinguish from which parameter the individual contribution comes. Nevertheless, in contrast to a deterministic approach, the user can see how uncertain the cINN is about the prediction due to its wide spreading of affected feature's prediction. In future contributions this can be handled by updating a joint density or stiffness variation, instead of individual features."*

Final technical corrections:
- Table 2: there are some entries written in green and some others in purple. Maybe, at least in the caption, this should be explained.
  *The table was changed, as colors are not permitted. All values meeting the threshold are depicted in bold. A remark is added to the caption.*
- "the authors pretend to identify parameters..." -> "The authors tried to identify parameters..."
  *Corrected.*
- "though high sensitivities do not directly promise high inverse predictions." -> "... do not directly promise highly accurate inverse prediction"
  *Corrected.*

*Dear Sarah Barber,*

*On behalf of all authors I gratefully thank you for reading the paper and providing valuable constructive criticism, which we believe has helped to develop and strengthen this work significantly. Please find our answers to all your comments, by either corrected or added text sections or comments on your suggestions/concerns. We hope to have correctly understood and adequately accommodated all your concerns. Thank you again for your contribution.*

**Specific comments**

1. INTRODUCTION

   - Line 52: can you quantify "computationally expensive" in terms of computational time as well as just number of iterations? How long does one iteration typically take?

   *This is impossible to quantify in a general manner, as it depends on the every model itself and the hardware you are using. The model generator we are using in this publication needs about: "…on average approx. 80 s on a single-core device." (As mentioned in Sec. 4.5) I will include this number as exemplary reference in the introduction:*

   *"Iterations are always model dependent, but as a reference for the real time consumption, the model generator used in this publication (Noever-Castelos et al., 2021a) takes on average approx. 80s on a single-core device for one iteration, i.e., model creation."*

   - Section 1.2: it would be better to introduce the three "problems" and then describe them, rather than describing one of them and then introducing the three problems.

   *This makes totally sense. Thank you! The order was swept, starting with the three bullet points introducing the issues and describing them afterwards.*

**2. SENSITIVITY ANALYSIS**

- Introduction: Usually one would expect the text at the start of a section before the first sub-section to introduce the section. Instead, you just talk about a previous paper, which is confusing. I would suggest inserting a proper introduction to the section here, and/or just moving the existing text into the first sub-section.

*As suggested, the section was moved into Sec. 2.1 and a proper introduction is added.*

- Section 2.1: Please explain briefly why you are using the Sobol method.
*"This method is widely used in research and is used here, as it also applies globally to non-linear models and analyzes interaction of input parameters on the model response."*
- Section 2.2: You refer to Figure 2 before Figure 1. Please swap the figures.
*Thanks for that notice. It was actually correct in latex, though the formatting process swapped them.*
- Section 2.2: Are you using one particular blade for this study or is it generalised? Please explain this better.

*Added:*

*"We will be performing the analysis on the DemoBlade of the SmartBlades2 project (SmartBlades2, 2016-2020)."*

- Line 111: With "In contrast to the simplified visualisation" do you mean the one used in the previous study?

*That was a bit confusing I have to admit. No, the figure shows a coarse mesh, however, in the anaylsis a more refined mesh is applied. I have adapted the sentence to:*

*"In contrast to this simplified visualisation in Fig 1..."*

- Line 118: Why "five equidistant nodes"?

*Added:*

*"The number of spline nodes can be chosen arbitrary; however, a high number increases the computational costs (more updating parameters) and can lead to collinear behavior if the nodes are to near, whereas a low number reduces the flexibility to adapt to short distance changes. For this study the number where chosen based on experience as a trade-off between computational costs and a sufficient approximation of a global parameter variation."*

- Line 163: "which does not necessarily improve the updating performance, but reduce the performance." This is a bit confusing. Does the second "performance" refer to the computational performance?

*Oh, this is really confusing. Yes, it is the computational performance. This sentence was changed to:*

*"This repeated information does not necessarily improve the updating results, but reduce the computational performance."*

**3. INN ARCHITECTURE**

- Lines 179-184: I would make the two colours in Fig. 4 more clear - it's hard to see them and differentiate between them.

*We have changed the color and the thickness of the lines, to make it more clear.*

- Line 190-198: can you give non-cINN-experts an idea of what the consequences of the flattening process are? I find myself not able to understand the effect of this on the results and it would be nice if you helped me out here (and others).

*There are no effects on the results, it is just about how the data is processed in the network:*

*"A consequence is, that the sub-networks cannot make use of convolutional layers, but have to include feed-forward layers. However, this will not have any significant impact on the result. As mentioned before, the conditions and input features are stacked in the sub-networks, which thus need a similar spacial shape. Consequently, the conditional network has to flatten the shape to a vector for each output, in order to agree with the input shape in the sub-networks."*

- Line 201: Please explain the table structure briefly. Remind us what the different clusters are.

*"As previously explained the conditional network processes the conditions c and has 5 outputs at different stages of the processing. Each of this outputs is fed into a cluster of 3 CCs. the configuration for each cluster and the corresponding hyperparameters for the conditional network, cINN and sub-networks is summarized in Table 3."*

**4. MODEL UPDATING**

- Line 244: Please quantify this, i.e. instead of "most of the values hit the ground truth." write something like "x% of the  values are within x% of the ground truth"

*"However, the overall posterior prediction in this example is very good, as approx. 70\% of the predictions are within a range of $\pm\,0.05$ (standardized scale) of the ground truth."*

- Line 249: You write "Thus, the ideal case would correlate to an exact line with a slope m = 1." (also with the intercept = 0?) - R^2 is not a measure of how close m is to one, but of how close the points are to the regression line y = mx + c (isn't it???). Please clarify this discrepancy and forgive me if I'm wrong.

*You are right. This has to be mentioned. However, the slope accuracy (to 1) and the $R^2$ value correlate in general for these results. This would only differ significantly if a*

*systematic error appears in the predictions. For this publication the slope m was added as additional information in Figure 9 and is now discussed briefly in the text:*

*"Approximately 70% of the selected features reach a very satisfying linear correlation with $R^2>0.9$, while showing a slope m of approx. 0.9 or higher. For the rest of the discussion we will be sticking with the $R^2$-value for the accuracy, as the slope accuracy correlates with the $R^2$-value."*

**- Line 327: why 5%?**

*This is arbitrary chosen on behalf of a maximum measurement error. For example, in the SmartBlades 2 Project the errors estimated for the strain gauges or the accelerometers were below 5%.*

**- Line 331: Please quantify the statement "most of the input features are predicted as accurate as with a clean output." (i.e. what do you mean by "most" and "as accurate"?)**

*We hope this satisfies your concerns:*

*"As visually confirmed in Fig. A1 the other features do not show a wider spread (orange) than the original values (blue) and therefore do not suffer from any accuracy loss."*

**- Lines 377-378: Quantify these two statements too!**

*"Again, all mean values are close to 1 (90% with MAC≥0.995), so an overall excellent updating performance can be stated. Single predictions lead to worse results, as depicted by the minimum value (4.3% of all have a MAC≤0.98), especially for the higher order modes, though the MAC value of less than 0.8 is only obtained for the $10^{th}$ eigenmode of the free-free configuration."*

**- Line 386: "The counteracting intrinsic model ambiguities cancel each other out". Could you explain this a bit more please?**

*"The counteracting intrinsic model ambiguities discussed in Sec. 4.2 cancel each other out, i.e., the overall shell laminate properties are correctly predicted, although the individual stiffness or density of the layers (Biax90 and Triax) are not predicted accurately. So the cINN still correctly captures the global model behavior with respect to mass and stiffness distribution."*

**- Line 390: Quantify this!**

*"The overall cINN updating performance is strikingly good, with on average 90% of the mode shapes showing a MAC≥0.995."*

- Line 396: It would be better to first mention this when introducing Sobol above (I already mentioned that you should expain why you chose the method), and then refer to it here.

*This is already mentioned in Sec. 2.2:*

*"SALib uses the quasi-random sampling with low-discrepancy sequences technique from Saltelli et al. (2008) for the sensitivity analysis. To compute the Sobol index, the algorithms require a variation of each input feature individually for each of the n samples, which results in a total sample size of $n_{total}$ (dim(x) + 2) = 79,360 to compute the $1^{st}$ order Sobol indices."*

**CONCLUSIONS**

- Lines 471-475: Please say something about how realistic the assumptions were. You say that it should now be applied to a real life application. This means you think that the assumptions you made in this work will impact the results. How and why?

*We hope that the following rephrasing and additions make it clear, what are the limitations and what still should be targeted in future research:*

*"The cINN proved to be extremely capable of performing an efficient model updating with a larger parameter space. The physical model complexity in form of a Tymoshenko finite element beam is already at the state of the art level applied in industry. However, to ensure that the cINN learns the complete inverted physical model, it is important that all possibly relevant parameters have to be varied, so that the cINN is trained for all circumstances of variations for the model updating. Therefore, ongoing and future investigations should bring this method to a real life application, where the parameter space will span more relevant aspects of blade manufacturing deviations, such as e.g., adhesive joints."*

**Technical corrections**

*Thank you for your technical corrections, which were all considered in the revision.*